METHODS AND RESOURCES

# Dynamic landscape of protein occupancy across the *Escherichia coli* chromosome

**Lydia Freddolino** [1,2]*, **Haley M. Amemiya** [2,3], **Thomas J. Goss**[1], **Saeed Tavazoie** [4,5,6]*

**1** Department of Biological Chemistry, University of Michigan Medical School, Ann Arbor, Michigan, United States of America, **2** Department of Computational Medicine and Bioinformatics, University of Michigan Medical School, Ann Arbor, Michigan, United States of America, **3** Cellular and Molecular Biology Program, University of Michigan Medical School, Ann Arbor, Michigan, United States of America, **4** Department of Biological Sciences, Columbia University, New York, New York, United States of America, **5** Department of Systems Biology, Columbia University, New York, New York, United States of America, **6** Department of Biochemistry and Molecular Biophysics, Columbia University, New York, New York, United States of America

* lydsf@umich.edu (LF); st2744@columbia.edu (ST)

## Abstract

Free-living bacteria adapt to environmental change by reprogramming gene expression through precise interactions of hundreds of DNA-binding proteins. A predictive understanding of bacterial physiology requires us to globally monitor all such protein–DNA interactions across a range of environmental and genetic perturbations. Here, we show that such global observations are possible using an optimized version of in vivo protein occupancy display technology (in vivo protein occupancy display—high resolution, IPOD-HR) and present a pilot application to *Escherichia coli*. We observe that the *E. coli* protein–DNA interactome organizes into 2 distinct prototypic features: (**1**) highly dynamic condition-dependent transcription factor (TF) occupancy; and (**2**) robust kilobase scale occupancy by nucleoid factors, forming silencing domains analogous to eukaryotic heterochromatin. We show that occupancy dynamics across a range of conditions can rapidly reveal the global transcriptional regulatory organization of a bacterium. Beyond discovery of previously hidden regulatory logic, we show that these observations can be utilized to computationally determine sequence specificity models for the majority of active TFs. Our study demonstrates that global observations of protein occupancy combined with statistical inference can rapidly and systematically reveal the transcriptional regulatory and structural features of a bacterial genome. This capacity is particularly crucial for non-model bacteria that are not amenable to routine genetic manipulation.

**Data Availability Statement:** All sequencing data and analysis related to the manuscript have been deposited at the Gene Expression Omnibus under accession GSE142291.

## Introduction

Transcriptional regulation plays a central role in establishing adaptive gene expression states. In bacteria, the dominant regulators are transcription factors (TFs) [1,2] and sigma factors, which direct the activity of RNA polymerase holoenzyme to a specific subset of promoters [3,4]. The phenotypic state of the bacterial cell is determined in large part by its transcriptional

**Funding:** This sutdy was funded by HHS | National Institutes of Health (NIH): ST R01-AI077562; by HHS | National Institutes of Health (NIH): PLF R00-GM097033 and by HHS | National Institutes of Health (NIH): PLF R35-GM128637. The funders had no role in study design, data collection and analysis, decision to publish, or preparation of the manuscript.

**Competing interests:** The authors have declared that no competing interests exist.

**Abbreviations:** ATAC-seq, assay for transposase-accessible chromatin using sequencing; ChIP-seq, chromatin immunoprecipitation sequencing; CWT, continuous wavelet transform; EPOD, extended protein occupancy domain; FAIRE, formaldehyde-assisted isolation of regulatory elements; FDR, false discovery rate; FWHM, full width at half maximum; GO, gene ontology; heEPOD, highly expressed extended protein occupancy domain; IPOD, in vivo protein occupancy display; IPOD-HR, in vivo protein occupancy display—high resolution; LC-MS/MS, liquid chromatography with tandem mass spectrometry; MNase-seq, micrococcal nuclease digestion with deep sequencing; MS, mass spectrometry; MS/MS, tandem mass spectrometry; NGS, next-generation sequencing; ORF, open reading frame; RDM, rich defined medium; RNA-seq, RNA sequencing; TF, transcription factor; TFBS, transcription factor binding site; TPM, transcripts per million; tsEPOD, transcriptionally silent extended protein occupancy domain; WT, wild-type.

regulatory state, which, in turn, is dictated by the binding pattern of TFs and sigma factors across the chromosome, likely in interplay with structural factors such as the local supercoiling state [5].

At present, however, our knowledge of the complete wiring of bacterial transcriptional regulatory networks remains insufficient to fully predict or design regulatory responses to arbitrary environmental conditions. The case of *Escherichia coli* serves as an illustrative case study: Due to its status as a preeminent model organism and important human pathogen, the *E. coli* transcriptional regulatory network has been an intense subject of investigation for several decades. As a result, researchers have obtained an increasingly comprehensive and detailed map of the binding specificities and physiological roles of transcriptional regulators in this organism [6]. However, roughly one quarter of the approximately 250 TFs in *E. coli* have no available binding or regulatory data [7], and many more are virtual unknowns in terms of the signals that might alter their regulatory activity. Likely as a result of this knowledge gap, Larsen and colleagues recently found that despite our broad knowledge of the potential regulatory targets of *E. coli* TFs, our ability to predict regulatory behavior on the basis of expression levels of TFs is no better than it would be for random networks. The authors attribute this partly to the fact that even when a TF is expressed, in many cases, it will not bind its targets in the absence of additional signals [8]. Furthermore, *E. coli* represents a best-case scenario in terms of our knowledge state for a bacterial transcriptional regulatory network and for most bacterial species current databases lag far behind.

Expanding our capability to predict, and ultimately design, bacterial regulatory responses will be critical for controlling bacterial pathogenesis and engineering synthetic microbes in biotechnology applications. Achieving such a complete predictive understanding, however, requires substantial additional information both on the binding sites of as-yet uncharacterized TFs and the actual physical occupancy of sites for both known and uncharacterized factors across conditions. Widely used methods such as chromatin immunoprecipitation sequencing (ChIP-seq) pose difficulties on both fronts: They demand a combinatorial explosion of experiments to study many TFs across a variety of conditions and require either an antibody against each TF of interest or genetic manipulation sufficient to add an epitope tag to each target TF.

In order to significantly advance our understanding of transcriptional network dynamics and chromosomal structure, we sought to monitor, in parallel, the occupancy states of all DNA-binding proteins across a set of genetic and environmental perturbations. We argue that such comprehensive observations are critical for defining the global modes of transcriptional regulation and determining the regulatory logic that underlies adaptive reprogramming of gene expression, particularly given the importance of combinatorial logic by many factors and sites in dictating transcriptional output [9]. In order to achieve our goal, we decided to employ the concept of in vivo protein occupancy display (IPOD) which we, in a previous proof-of-concept study, demonstrated to reveal global occupancy of protein binding sites across the *E. coli* chromosome [10]. However, we had to introduce critical modifications and enhancements in order to deconvolve distinct contributions from sequence-specific TFs and RNA polymerase and define binding sites at high resolution. We will refer to this second-generation IPOD technology as in vivo protein occupancy display—high resolution (IPOD-HR). IPOD-HR enables efficient coverage of a large range of physiological conditions in relatively few experiments (1 experiment per condition, rather than the 1 experiment per TF per condition that would be required for ChIP-seq). As we demonstrate below, a single IPOD-HR experiment can reveal the occupancy dynamics of dozens of known and novel active TFs genome-wide, permitting rapid profiling of global transcriptional regulatory logic across different conditions. Furthermore, the comprehensive nature of IPOD-HR profiles enables efficient statistical inference of sequence specificity models (transcription factor binding site [TFBS] motifs) for active

TFs, both recapitulating well-known regulatory logic and revealing the presence and condition-dependent activities of novel regulatory elements.

Here, we characterized the dynamics of the global protein–DNA interactome of *E. coli* across a range of 3 physiological conditions and 3 genetic perturbations. Our observations allowed us to infer, in parallel, the activities of most annotated TFs across conditions, and provided a catalog of many additional likely regulatory sites and DNA sequence motifs for uncharacterized TFs. With the compact set of experiments, we reveal the dramatic regulatory dynamics of dozens of TFs that collectively shape the response of *E. coli* to changing environments. In sharp contrast, we find that at the kilobase scale, the genome is characterized by a set of relatively static structural domains, which consist of transcriptionally silent loci with dense protein occupancy that appear mostly constitutive across a range of physiological conditions. These regions, which we refer to as extended protein occupancy domains (EPODs) following the nomenclature of Vora and colleagues [10], appear to act, at least partially, to suppress prophages and mobile genetic elements.

Because our approach does not rely on prior knowledge of TFs of interest or genetic manipulation of the target organism, but rather only on essential physicochemical properties of protein–DNA complexes, we expect that it will be broadly applicable across bacterial species, even those which cannot be cultured or genetically manipulated. Our approach lays the technical and analytic foundation to rapidly characterize the regulatory and structural features of any bacterial chromosomes.

## Results

### Global high-resolution profiling of condition-dependent transcription factor occupancy across the *E. coli* chromosome

The IPOD-HR procedure is shown in schematic form in **Fig 1A**: Cells are grown under a physiological condition of interest, fixed using formaldehyde, and then lysed. Heavy digestion of the chromosomal DNA provides minimized DNA fragments that may be in either a protein bound or unbound state. The protein bound DNA fragments are subsequently isolated using a phenol–chloroform extraction. Under appropriate buffer conditions, the amphipathic protein–DNA complexes are depleted from the aqueous phase and partition to a robust disc at the aqueous–organic interface [10].

As we will demonstrate below, the measurements enabled by IPOD-HR can subsequently be used for a broad range of downstream analyses, such as simultaneous monitoring of the activities of characterized TFs, large-scale inference of binding motifs for previously uncharacterized DNA-binding proteins, and identification of key occupancy sites driving previously unrecognized gene regulatory logic. To accomplish these objectives, it is essential to separate out the occupancy signal of RNA polymerase from that of specific regulatory factors of interest. Otherwise, the strong occupancy signal caused by RNA polymerase could mask changes in protein occupancy that in fact provide regulatory information. To deconvolve occupancy caused by sequence-specific TFs and that of RNA polymerase, we subtract the normalized RNA polymerase ChIP-seq signal from that of the normalized raw IPOD-HR signal (see Methods for details), generating a corrected IPOD-HR profile that is a more precise representation of the cell's dynamic regulatory state (**Fig 1B**). IPOD-HR is conceptually similar to the original IPOD method [10] in terms of overall workflow, but contains critical optimizations and extensions designed to permit genome-wide identification of binding by TFs and organizing factors such as nucleoid-associated proteins in a condition-specific manner. On the experimental end, IPOD-HR incorporates a more stringent washing procedure (the inclusion of a Tris base wash and an additional interphase extraction step to increase the specificity of isolation of protein–

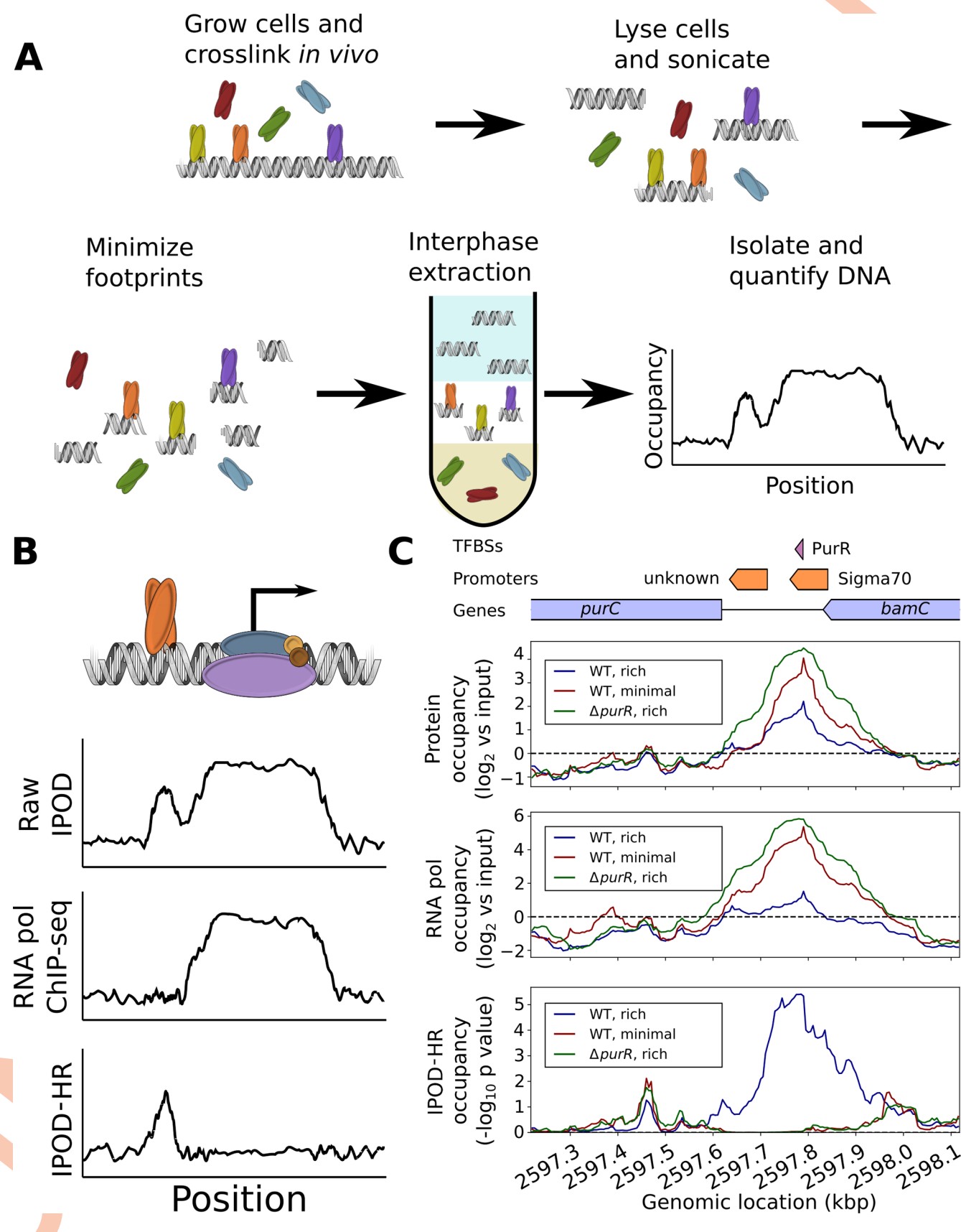

**Fig 1. Schematic of IPOD-HR technology and detection of context-dependent binding by TF PurR. (A)** Overall workflow for isolation of the IPOD-HR fraction and quantification of total protein occupancy. **(B)** The final IPOD-HR signal is obtained by subtracting a normalized RNA polymerase occupancy signal from the raw IPOD-HR protein occupancy, resulting in a polymerase-corrected signal. **(C)** Example of RNA polymerase–corrected IPOD-HR profile upstream of the *purC* gene, where subtraction of RNA polymerase occupancy from the raw IPOD-HR signal properly reveals a PurR binding site in rich media that is lost upon deletion of *purR* or transition to minimal media. In the schematic above the plots, blue regions show genes, orange regions show promoters, and purple regions show annotated TFBSs. ChIP-seq, chromatin immunoprecipitation sequencing; IPOD, in vivo protein occupancy display; IPOD-HR, in vivo protein occupancy display—high resolution; TFBS, transcription factor binding site; WT, wild-type.

DNA complexes), as well as pretreatment of the cells with rifampin prior to cross-linking to minimize the contributions of RNA polymerase occupancy to the observed signal, and an RNA polymerase ChIP-seq experiment in parallel to allow separation of RNA polymerase occupancy from that of other proteins (more information on the interplay of RNA polymerase occupancy and that of other proteins is given in **S1 Text**). On the computational end, IPOD-HR makes use of a completely rewritten analysis pipeline (as detailed in Methods) that provides proper adjustment for RNA polymerase occupancy, gives an integrated workflow for calculation of total protein occupancy (including uncertainty estimates), and allows for the identification of key occupancy features such as occupancy peaks corresponding to individual binding sites, as well as large regions of high protein occupancy that act as silencing complexes on the *E. coli* genome.

We note in passing that, at first glance, IPOD may seem to share superficial similarities with formaldehyde-assisted isolation of regulatory elements (FAIRE, originally described in [11]). However, FAIRE experiments were designed to detect regions of nucleosome-depleted DNA in eukaryotic chromosomes. IPOD was independently developed to detect occupancy of individual factors in prokaryotic chromosomes [10], and IPOD-HR contains further optimizations and additional experimental and computational steps to improve performance in detecting both localized and large-scale protein occupancy in bacteria.

An illustrative example of the ability of IPOD-HR to identify regulatory protein occupancy, its dynamics across conditions, and the importance of factoring out the RNA polymerase signal is shown in **Fig 1C**. We consider the IPOD-HR occupancy profiles for the promoter region upstream of the *purC* gene in wild-type (WT) and Δ*purR* cells during growth in rich defined medium (RDM). Based on the characterized behavior of PurR (which binds DNA in response to exogenous purine supplementation [12,13]), under this growth condition, transcription of *purC* should be repressed by binding of PurR to its promoter. However, if one considers only the raw IPOD-HR occupancy profiles (top panel), binding to the PurR site in this region is apparent in both WT and Δ*purR* cells. The resolution to this seeming paradox becomes apparent through inclusion of the correction for RNA polymerase occupancy (middle panel), which is substantially higher in Δ*purR* cells. As expected, the resulting corrected IPOD-HR occupancy profiles (bottom panel) reveal a protein occupancy peak directly on top of the annotated PurR binding site in this region in the WT cells, and no detectable occupancy in the Δ*purR* cells. This demonstrates the ability of IPOD-HR to reveal condition-dependent TF occupancy dynamics even in regions that may overlap with RNA polymerase binding. In the following sections, IPOD-HR refers to the RNA polymerase–corrected occupancy signal, unless otherwise noted.

## Local and large-scale protein occupancy patterns across the *E. coli* chromosome

To benchmark our ability to quantitatively profile protein occupancy at high spatial resolution, we performed IPOD-HR on *E. coli* cells from mid-exponential growth in RDM (**Fig 2A**). Over the length of the chromosome, we observed a large number of small peaks, presumably

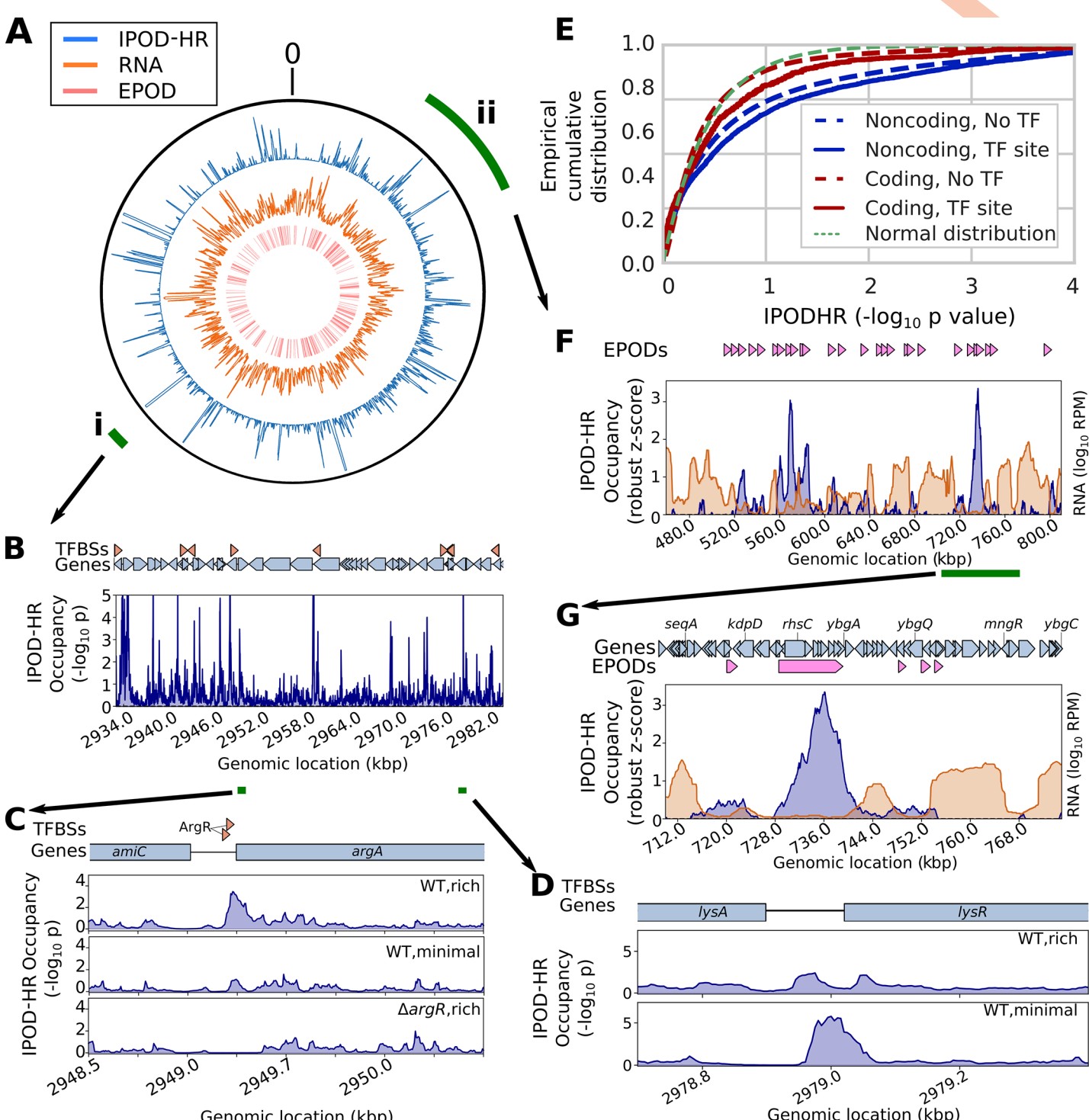

**Fig 2. IPOD-HR profiles reveal rich high-resolution occupancy dynamics and large-scale structural features across the chromosome.** (**A**) Outer track: IPOD-HR occupancy (robust Z-scores, 5-kb moving average); middle track: total RNA read density (5-kb moving average); inner track: locations of inferred EPODs. The outer green wedges mark the portion of the chromosome shown in subsequent panels. The origin of the coordinate system is oriented at the top of the plot. All data in this figure are for the "WT,rich" condition unless otherwise noted. (**B**) IPOD-HR occupancy measured during growth in glucose RDM, in the vicinity of wedge **i** from panel **A**. Green segments below the genomic coordinates indicate the regions highlighted in panels **C–D**. (**C**) Condition-dependent occupancy changes at the ArgR binding sites upstream of *argA*. (**D**) Identification of condition-specific occupancy of a likely LysR binding site between *lysA* and *lysR*. (**E**) Cumulative histograms showing RNA polymerase ChIP-subtracted IPOD-HR occupancy in coding vs. noncoding regions and at sites that match known TFBSs from RegulonDB [7], compared with the curve that would be expected from a standard normal distribution of scores. Additional descriptive statistics and significance calls are given in **S1 Table**. (**F**) Occupancy (blue)

and total RNA abundance (orange) for a selected sector of the genome (wedge **ii** from panel **A**), showing the presence of several EPODs in regions corresponding to low RNA abundance; rolling medians over a 5-kb window are plotted, with RNA read densities shown in units of RPM. **(G)** Magnification of the region highlighted by the green bar in panel **F**, illustrating a silenced region in and around *rhsC*, alongside flanking areas of low IPOD-HR occupancy and high transcription. A 5-kb rolling median is plotted. ChIP, chromatin immunoprecipitation; EPOD, extended protein occupancy domain; IPOD-HR, in vivo protein occupancy display—high resolution; RDM, rich defined medium; RPM, reads per million; TF, transcription factor; TFBS, transcription factor binding site; WT, wild-type.

corresponding to protein binding events at individual regulatory sites. In addition, we observed many large-scale (>1 kb) regions of high occupancy, which we refer to as EPODs, following the nomenclature of Vora and colleagues [10]. An example of condition-dependent changes in binding of local TFs is shown in **Fig 2B and 2C**. Examination of an approximately 50-kb slice of the genome reveals dozens of small occupancy peaks, with a visually apparent enrichment in intergenic regions (**Fig 2B**). Many such peaks, which presumably correspond to individual protein binding events, coincide with known TFBSs. For example, the region upstream of *argA* (**Fig 2C**) shows strong occupancy at known ArgR binding sites and condition-appropriate occupancy dynamics including weakening of binding in arginine-poor conditions [14,15] and loss of occupancy upon deletion of the *argR* gene. At the same time, similar occupancy patterns can be observed at many sites lacking an annotated TFBS, as seen in **Fig 2D**, where a conditionally dynamic binding site is apparent between *lysA* and *lysR* and a constitutive binding site in the *lysA* gene body. This dynamic site between the genes likely corresponds to binding of LysR itself in our minimal media condition, as LysR is known to bind somewhere in the region just upstream of *lysR* to repress transcription of *lysR* and activate transcription of *lysA* [16], but the precise location of the binding site has not previously been determined to our knowledge, and thus its exact coordinates are not present in common databases such as RegulonDB [7]. To facilitate inspection of other regions of interest, we provide complete occupancy traces for the conditions studied in **S1 Data**. As expected, at a genome-wide scale IPOD-HR signals show both significantly higher occupancy in intergenic regions relative to coding regions and significantly higher occupancy at annotated TFBSs relative to other regions of the chromosome (**Fig 2E**; see also **S1 Table**), demonstrating a strong overlap of the observed protein occupancy with transcriptional regulatory sites. Indeed, applying peak calling to the IPOD-HR signal demonstrates an increasingly strong overlap with known TFBSs as the threshold for peak calling is increased (**S1 Fig**; a full listing of peak calls is given in **S2 Data**).

It is also apparent by inspection of the genome-wide occupancy shown in **Fig 2A** that many extended regions of high protein occupancy coincide with regions of relatively low transcription. For example, in **Fig 2F**, we show a typical approximately 300-kb region with alternating segments of high protein occupancy that have relatively low transcription, with those of low protein occupancy and relatively high transcription (also apparent in the higher-resolution plot in **Fig 2G**). Thus, in addition to revealing occupancy at the level of individual regulatory sites, IPOD-HR enables tracking of the behavior of large, densely protein occupied regions of the chromosome that appear to coincide with transcriptionally silent loci. We will explore both of these prototypic classes of occupancy, in more detail, below.

## Transcription factor and sigma factor occupancy dynamics across genetic and environmental perturbations

Since IPOD-HR occupancy profiles show highly enriched overlaps with known TFBSs (**Fig 2E**), we asked whether IPOD-HR profiles can be used to reveal the occupancy dynamics for known *E. coli* TFs across a set of conditions. Indeed, we find that IPOD-HR reveals consistent and condition-appropriate regulatory logic at the level of individual regulons and patterns of

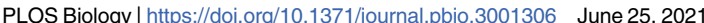

regulatory behavior across regulons. As expected, strains with each of 3 single TF deletions (*argR*, *lexA*, and *purR*) show global loss of occupancy at the ensemble of annotated sites for the corresponding TFs (**Fig 3A**) relative to growth under equivalent conditions of WT cells (WT, rich). Analysis of condition-dependent changes in the occupancy of binding sites for single TFs likewise recapitulates expected behavior; for example, ArgR [17] and PurR [12,18] show enhanced binding to DNA in the presence of amino acid or nucleobase ligands that are supplied directly in our rich media conditions, and the IPOD-HR occupancy signal shows global loss of occupancy for binding sites of both of these TFs in nutrient-depleted conditions (minimal media) when compared with exponential growth in rich media (**Fig 3A**). In contrast, RutR shows increased overall occupancy in minimal media relative to rich media conditions, consistent with the known inhibition of RutR binding by thymine and uracil [19]; similar behavior is observed for the cytidine-responsive CytR [20,21]. Sites for ArcA, which becomes active under low oxygen conditions [22,23], show strong increases in occupancy in stationary phase at high culture density (**Fig 3A**). Taken together, these findings demonstrate that IPOD-HR provides a quantitative readout of changes in regulatory activity across a broad range of physiological and genetic perturbations.

The utility of IPOD-HR in tracking the occupancy of known binding sites extends to global regulators with large characterized regulons; for example, comparison of the occupancy at annotated Lrp binding sites observed in IPOD-HR data to the occupancy of Lrp itself at the same sites in Lrp ChIP-seq data under similar conditions shows a significantly more similar occupancy profile than is seen for other TFs with large regulons (**Fig 3B**), as would be expected if the IPOD-HR profile were tracking Lrp occupancy across conditions. Importantly, this concordance is maintained even in the presence of many other occupancy changes being measured for other proteins at locations throughout the genome, and potential interference from other factors binding to sites overlapping the Lrp sites.

Similar insights can be obtained across the *E. coli* transcriptional regulatory network by extending our analysis to all characterized TFBSs annotated in RegulonDB [7]; we show the resulting condition-dependent occupancies in panel A of **S2 Fig**. As is apparent from the raw occupancies, *E. coli* TFs differ substantially from each other in the strength of the IPOD-HR footprint that they generate (leading, for example, to the very strong signals for factors such as IdnR). Across the conditions in our study, 122 out of the 176 TFs in RegulonDB had at least 1 site with detectable occupancy (robust z-score >3), and thus we are able to provide data on the characterized binding sites for the majority of TFs in *E. coli*. The remaining TFs likely represent a combination of factors that are not expressed or active under the limited set of conditions that we considered and those that cannot be cross-linked efficiently to DNA by formaldehyde (e.g., Lac repressor, which was previously shown not to cross-link effectively to DNA with formaldehyde [26], and does not show substantial IPOD-HR signal) or do not partition appropriately in the phenol–chloroform extraction. For the majority of factors showing detectable occupancy, condition-dependent dynamics can be observed most clearly by normalizing the TF level occupancy by the highest occupancy condition observed for that factor (**S2 Fig**, panel B).

By applying an unsupervised clustering approach (see Methods for details), we identified transcriptional regulatory modules that show consistent co-regulation across the conditions in our study. We found clustering of TFs with highly similar behavior (**Fig 3C**) that coordinate, for example, oxidative metabolism (red), amino acid uptake and synthesis (brown), and iron homeostasis (green). We also observe several cases where related regulators or regulatory cascades are clustered together; for example, the well-known regulators of carbon metabolism Cra and CRP (blue), NtrC and its transcriptional activator Fis (purple), or the tightly intertwined acid response regulators GadE, GadW, and GadX (pink). In order to test whether the

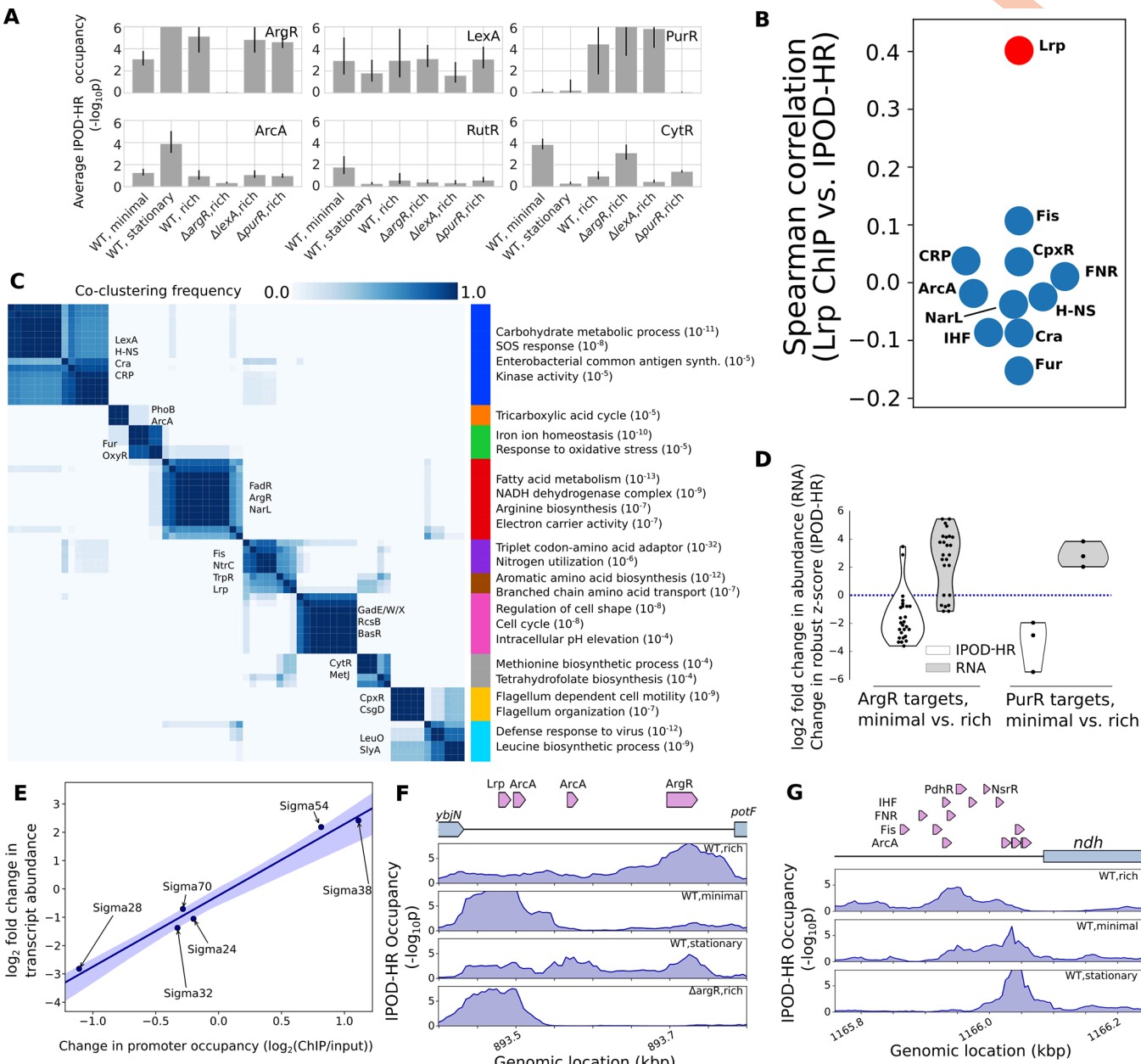

**Fig 3. IPOD-HR profiles reveal global binding activity of known TFs and sigma factors. (A)** Average (geometric mean) occupancies for all annotated binding sites of the 6 indicated TFs under each indicated condition. Error bars indicate a 95% confidence interval based on parametric bootstrapping with pessimistic assumptions; see Methods for details. The number of detectable sites used to estimate the condition-specific occupancies were 30, 10, 2, 45, 6, and 9 for ArgR, LexA, PurR, ArcA, RutR, and CytR, respectively. **(B)** Spearman correlations between all occupancy values at annotated binding sites for the indicated TFs (all TFs with at least 50 sites in the RegulonDB database) in the IPOD-HR vs. Lrp ChIP data sets. Points shown in red have a statistically significant correlation (FDR-corrected *p*-value < 0.05). Annotated binding sites are from RegulonDB release 9.4, prior to inclusion of the ChIP data used here, with overlapping or bookended sites for the same TF merged prior to analysis); data are from [24] (Lrp ChIP) or the present study (IPOD-HR). Data are taken from the most closely equivalent conditions (log phase growth in minimal media, log phase growth in rich media, and stationary phase in rich media), although the carbon source is different (glycerol vs. glucose). **(C)** Heat map showing the consensus clustering (co-occurrence frequencies) of the pattern of occupancy dynamics for the regulons of all considered TFs across the varied nutrient conditions in this study (see Methods for details). Consensus division into 10 clusters via agglomerative clustering is shown at right; for each cluster, representative TFs (on matrix) and regulated GO terms (right) are shown, with numbers in parentheses indicating the approximate *p*-value for enrichment of that GO term. A full listing of *p*-values is given in **S2 Table**. **(D)** Changes in occupancy and target gene transcript level for all annotated repressive binding sites of ArgR and PurR (for minimal media vs. rich media), in each case demonstrating the strong and oppositely directed changes in binding and regulatory effects across the regulons. **(E)** Correlation of promoter-level occupancy changes (measured by RNA polymerase ChIP-seq) and changes in transcript abundance, shown for the WT stationary phase condition compared with exponential phase. Shaded area shows a bootstrap-based 95% confidence interval. **(F)** IPOD-HR protein occupancy profiles in the vicinity of the *potF* promoter under the indicated

conditions. Drawn TFBSs are taken from Ecocyc [25] reflecting recent updates in known TFBSs in this region. **(G)** IPOD-HR occupancy profiles upstream of *ndh* under the indicated conditions. For all rows of TFBSs except the top, all TFBSs in a given row correspond to the factor named at the beginning of that row. ChIP, chromatin immunoprecipitation; ChIP-seq, chromatin immunoprecipitation sequencing; FDR, false discovery rate; GO, gene ontology; IPOD-HR, in vivo protein occupancy display—high resolution; TF, transcription factor; TFBS, transcription factor binding site; WT, wild-type.

co-clustering more generally indicated a significant overlap in the regulons of factors, or merely that they respond similarly to changing conditions, we calculated the pairwise Jaccard indices (i.e., the size of the intersection of regulons between 2 TFs divided by the size of their unions, in terms of number of regulated genes) for all pairs of TFs considered in our analysis. We then compared the distribution of Jaccard indices for TFs that came from the same cluster to that of Jaccard indices for pairs of TFs that did not co-cluster in **Fig 3C**. TF pairs that co-clustered were almost equally likely to have a nonzero Jaccard index (and thus share at least some regulatory targets) than were TF pairs from different clusters (odds ratio 0.95, $p = 0.76$, Fisher exact test). Thus, co-clustering appears to reflect different TFs that have similar condition-dependent occupancy profiles, rather than factors that regulate identical targets (likely minimizing redundancy of regulatory information in favor of combinatorial regulation of a single target by factors that each sense different stimuli). We expect that substantially more insight into these patterns may be obtained in the future through application of IPOD-HR to a broader array of conditions.

We thus find that IPOD-HR occupancy profiles can provide detailed, site-level, condition-specific information on regulatory protein occupancy across the entire chromosome. By comparing changes in protein occupancy with changes in transcript levels across conditions, we can relate changes in protein occupancy to their positive or negative regulatory consequences. This can be seen for 2 nutrient-sensing transcriptional repressors with sites annotated in RegulonDB, ArgR, and PurR, across changes in nutrient conditions (**Fig 3D**). The changes in protein occupancy and regulatory output at these sites show that for both factors, there is a strong loss of protein occupancy at repressive ArgR and PurR sites and a corresponding increase in transcriptional output in regulated genes, when considering minimal media relative to rich media.

Since each IPOD-HR global protein occupancy data set is performed alongside an RNA polymerase ChIP-seq experiment, we can easily track promoter occupancy alongside TFBS occupancy. The use of rifampin permits transcriptional initiation, but prevents elongation past a few nucleotides [27]. Thus, these data sets are ideal for identifying regulation at the level of RNA polymerase (e.g., via different sigma factors). The differential patterns of RNA polymerase occupancy show strong correlations with transcript levels for each sigma factor's regulon across a range of conditions. As shown in **Fig 3E**, when comparing logarithmic versus stationary phase conditions, the changes in transcript abundance and RNA polymerase promoter occupancy show a Spearman correlation of 0.94 ($p = 0.005$); an equivalent comparison for changes in occupancy versus expression for cells grown in minimal media yields similar results (Spearman correlation 0.77, $p = 0.072$).

In addition to revealing the occupancy of individual, well-separated TFBSs, IPOD-HR occupancy can also yield insight into TF behavior at regulatory regions bound by multiple factors. An example is shown in **Fig 3F** for the *potF* promoter, which contains closely spaced Lrp, ArcA, and ArgR binding sites. The observed occupancy profile appears decomposable into a contribution from Lrp binding (strong in minimal media and moderate in the RDM conditions), ArgR binding (strongest in rich media, moderate in stationary phase RDM and in minimal media, and absent in Δ*argR* cells), and ArcA binding (mainly in stationary phase). Binding of the Lrp site at the *potF* promoter is known to be strongest in minimal media and

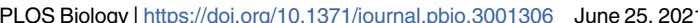

weaker but still present in RDM (both in exponential and stationary phase; see Fig 4A of [24]); the broad occupancy around the annotated Lrp site is likely due to weaker binding of other sites from the Lrp octamer. The inferred behaviors of ArgR and ArcA binding are likewise consistent with the overall behavior of sites for those TFs in our experiments (**Fig 3A, S2 Fig**). A more complex example can be seen in **Fig 3G**, where we show the *ndh* promoter containing overlapping binding sites for 6 different TFs. At such a complex promoter, deconvolution of the observed occupancy is still possible subject to consistency assumptions (that sites for a given TF will all change occupancy in the same direction in unison, even if by different amounts). Based on this assumption, the predominant contributors to binding at the *ndh* promoter appear likely to be IHF and PdhR in the "WT,rich" condition, IHF and NsrR in the "WT,minimal" condition, and NsrR and ArcA in the "WT,stationary" condition. However, assignments of occupancy to specific factors at such a complex promoter cannot be made with certainty on the basis of IPOD-HR data alone; combination with ChIP data under similar conditions (as invoked in the discussion above for **Fig 3F**) can aid in assignment, as could more formal methods for deconvolving the protein occupancy signal based on the overall distribution of sites for each factor across the genome (similar to the approach taken by CENTIPEDE for eukaryotic chromatin accessibility data [28]); development of such analysis tools is an area of active research.

## Global occupancy dynamics reveals the action of new DNA-binding proteins

Despite extensive annotation efforts, at present, fewer than 1,100 of the 3,560 annotated transcriptional units present in the RegulonDB database have any annotated regulation by TFs assigned to them [7]. While several recent notable efforts have sought to expand the completeness of these regulatory annotations by studying the DNA-binding preferences of purified TFs [6,29,30], or via computational inference of likely additional regulation [31] and regulatory modules [32], none of these methods provides either direct evidence for binding in vivo or information on condition-dependent changes in occupancy. IPOD-HR, in contrast, can provide both. Furthermore, the protein occupancy signals thus obtained provide information on occupancy of both well-characterized and uncharacterized proteins. In fact, many dynamic IPOD-HR peaks occur in promoters with no previous annotation for TFBSs, as we will discuss in detail in the following section.

   A representative example of an orphan occupancy peak is seen upstream of the gene *sdaC* (**Fig 4**). In our RNA sequencing (RNA-seq) data, *sdaC* transcript levels are nearly 20-fold higher during exponential growth in rich media (317.3 transcripts per million [TPM]) compared with either exponential growth in minimal media (17.9 TPM) or stationary phase in rich media (16.7 TPM). IPOD-HR occupancy profiles (**Fig 4A**) show a likely transcriptional activator binding site upstream of the *sdaC* core promoter, which shows strong occupancy in the WT M9/RDM/glu conditions but not the related conditions where *sdaC* expression is lower; in contrast, the only annotated TFBS in that region is a repressive Lrp site 200 bp downstream of the occupancy peak. To identify the TF(s) responsible for that occupancy, we used a biotinylated bait DNA matching the sequence of the *sdaC* promoter region to isolate proteins bound to that region from *E. coli* cells grown in the WT M9/RDM/glu condition (**Fig 4B**). Mass spectrometry on isolated bait-dependent bands revealed 2 poorly characterized TFs, UlaR and YieP, which showed highly enriched binding to the *sdaC* promoter (see **S3 Table**). While UlaR proved difficult to purify due to poor solubility and was thus excluded from further analysis, we found that purified YieP does indeed show specific shifting of the *sdaC* promoter in an electrophoretic mobility shift assay (**Fig 4C**). Consistently, recent RNA-seq data on a Δ*yieP* strain

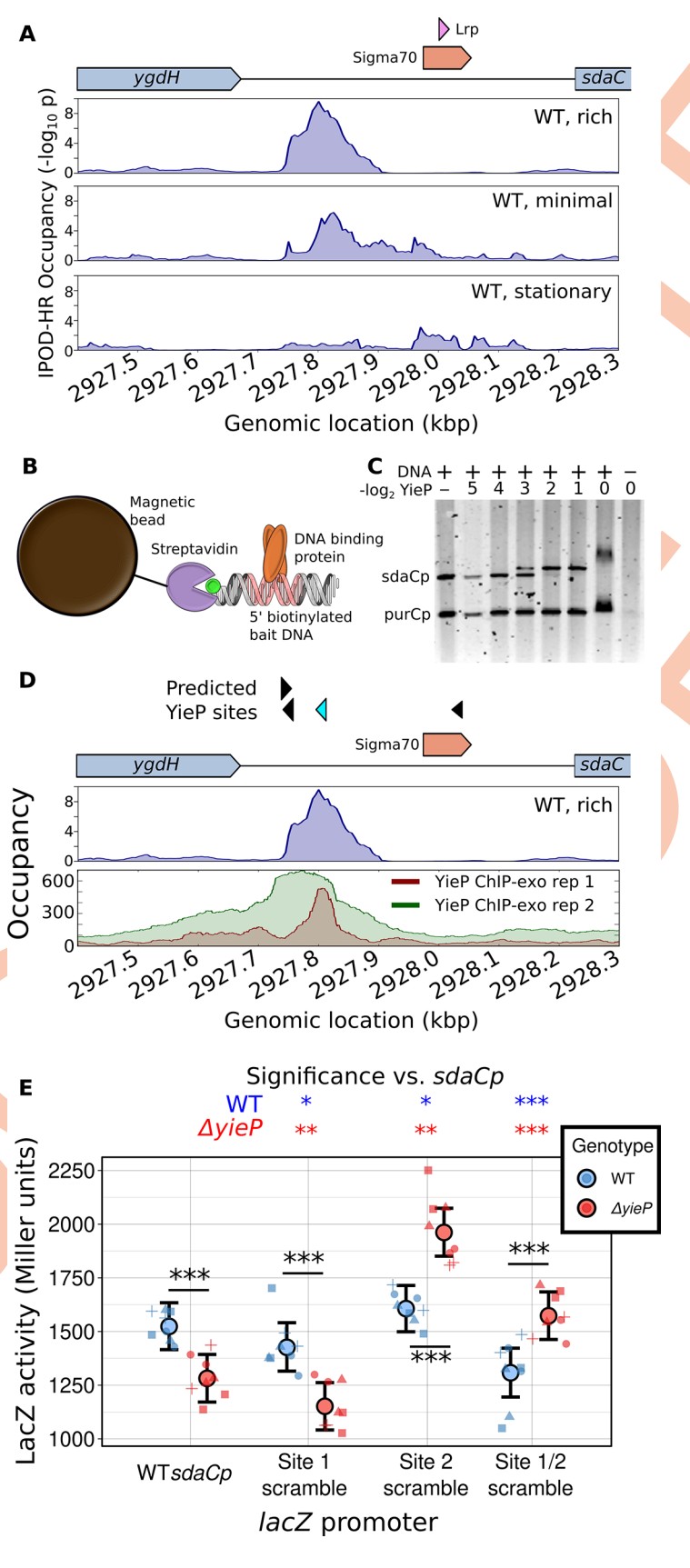

**Fig 4. Experimental identification of the protein bound to a novel occupancy peak upstream of the *sdaC* promoter.** **(A)** IPOD-HR profiles upstream of *sdaC* in rich (M9/RDM/glu) media, minimal (M9/glu) media, and in rich media in stationary phase (the drawn Lrp binding site is taken from Ecocyc [25] and is not present in RegulonDB). **(B)** Schematic of pulldown/mass spectrometry experiments used to identify factors binding the *sdaC* promoter. **(C)** Gel shift experiments showing specific interaction of YieP with the *sdaC* promoter. Increasing concentrations of purified His$_6$-YieP are incubated with a mixture of fluorescein-labeled promoter regions from *sdaC* and *purC* and then run on a gel, demonstrating specific shifting of the *sdaC* promoter region. YieP concentrations are given as the number of 2-fold dilutions relative to full strength. **(D)** Comparison of IPOD-HR occupancy profile (as in panel **A**) with ChIP-exo data from [31], with the latter given as total read counts (parsed from GEO accession numbers GSM3022131 and GSM3022132). The top track of predicted YieP sites shows significant hits for the YieP motif identified based on that ChIP-exo data set. Out of 1,025 potential YieP sites in the genome, the location highlighted in cyan is tied for 10th highest score (identified using FIMO; see Methods for details). Occupancy signal is given as $-\log_{10}(p)$ for the IPOD-HR track or raw counts (averaged across strands) for the ChIP-exo tracks. **(E)** Results of Miller assays in which *lacZ* transcription is driven by a copy of the sdaC promoter, either with the native sequence (WT) or with one or both of the apparent YieP binding sites scrambled, in both a WT and *yieP* background. Large points and error bars show a posterior mean and 95% credible interval from a Bayesian analysis; small points show individual data points, with symbols denoting the day on which data were gathered (a total of 8 biological replicates split across 4 different days were performed for each strain). Significance is assessed using 1-sided Bayes factors with the interpretive scale of Kass and Raftery [33] (\*: Substantial, \*\*: Strong, \*\*\*, Decisive). Stars within the plot denote direct comparisons of the WT and *yieP* strains for each promoter, whereas those above the plot denote comparisons of each promoter variant with the original within a given genetic background. ChIP, chromatin immunoprecipitation; IPOD-HR, in vivo protein occupancy display—high resolution; WT, wild-type.

shows a significant drop in *sdaC* transcript levels (2.7-fold change, q = 7.6 \* 10$^{-18}$) relative to isogenic cells with a plasmid-born reintroduction of YieP during growth in LB media (personal communication, C. Bianco and C. Vanderpool).

YieP was recently (and independently) selected by Palsson and colleagues as a validation case to be used in their consideration of computational methods for identifying the binding sites of orphan TFs and subjected to ChIP-exo analysis on cells grown in glucose minimal media using epitope-tagged YieP [31]. Indeed, their data demonstrate both strong direct YieP occupancy, and a high confidence YieP motif match, at the precise position of the occupancy peak detected in our IPOD-HR data set (Fig 4D). Based on the relative intensity at that position across conditions, combined with the expression data noted above, we infer that YieP binds to the *sdaC* promoter in nutrient-replete conditions and acts as a transcriptional activator (explaining the solitary strong peak in our "WT,rich" condition), whereas in other conditions, YieP binding is weakened (but not abolished), and additional factors such as Lrp likely bind downstream of the YieP site to repress *sdaC* transcription.

We must emphasize that the discovery of YieP binding sites through IPOD-HR and subsequent mass spectrometry experiments (by us) occurred in parallel with the ChIP-exo experiments of Gao and colleagues, and, indeed, represent highly complementary paths for identification of the binding sites for orphan TFs, with one centered on a candidate protein and the other on candidate sites.

In order to directly assess the regulatory role of the identified YieP binding sites, we integrated a *lacZ* reporter at the *araBAD-araC* locus in both WT and Δ*yieP* backgrounds, driven by variants of the *sdaC* promoter that either match the original sequence, or have 1 or both YieP binding sites scrambled (see Methods for details). Consistent with the RNA-seq data described above, we observed that deletion of *yieP* leads to a significant drop in reporter activity with the WT *sdaC* promoter (Fig 4E, "WT *sdaCp*"). Interpretation of the single-site scrambles is complicated by the fact that the promoter variants appear to have substantively different effects in the WT and *yieP* backgrounds; however, what is clear is that removal of both YieP binding sites (Fig 4E, "Site 1/2 scramble") leads to a drop in reporter activity in the WT background that is virtually identical to that observed with deletion of *yieP*, whereas in a *yieP* background, there is instead a significant increase in reporter activity when the sites are scrambled.

The sign epistasis observed between *yieP* genotype and the presence of YieP sites at the *sdaC* promoter argues that the observed YieP binding sites are functional regulatory sites, but that YieP also acts indirectly at this promoter (either through alterations of cellular physiology in the *yieP* background or genetic interaction with another regulator acting at this promoter, e.g., by repressing and/or directly competing for binding with a repressor that acts here).

The example presented here of regulation of *sdaC* by the uncharacterized TF YieP high-lights the broad potential for using IPOD-HR to rapidly identify and characterize previously cryptic regulatory connections. IPOD-HR thus complements the multitude of other approaches noted above (based on, e.g., promoter libraries or computational inference) and provides the unique benefit of directly assessing binding to DNA in vivo, at native loci, under physiological conditions of interest.

The utility of IPOD-HR in identifying the activity of previously uncharacterized TFs moti-vates its extension to a genome-wide scale, providing an in vivo complement to high-through-put in vitro screening methods such as genomic SELEX [6]. By applying peak calling to our IPOD-HR data sets across the 6 conditions considered in the present study, we were able to identify thousands of likely TFBSs, many of which are not identifiable based on existing data-bases. A comprehensive listing of peak calls across conditions and thresholds is given in **S2 Data**. To compare the peak sets identified from IPOD-HR data with our existing state of knowledge, we divided the peak calls obtained from IPOD-HR into a set of annotated TFBSs from RegulonDB and a set of binding sites predicted using all known PWMs available in the SwissRegulon database (see Methods for details). We find that across a range of thresholds, approximately half of the binding sites identified by IPOD-HR overlap with either known or predicted sites, whereas the other half represent novel binding sites which likely (as in the case of the YieP site described above) reflect the activity of poorly annotated or orphan TFs. Pooling the newly identified binding sites across conditions, our IPOD-HR data sets are able to provide a total of 14,271 putative TFBSs, representing 5,090 unique sites, which are occupied in vivo under at least 1 condition; we also track the dynamics of occupancy of those sites across condi-tions. This extensive map of chromosomal occupancy and its dynamics provide the commu-nity with a wealth of known and putative novel regulatory interactions that can be further explored and validated by follow-up experiments such as those shown in **Fig 4**. It is also possi-ble to cross-reference the newly identified binding sites with high-throughput studies of poten-tial binding locations for newly characterized TFs (e.g., ChIP or genomic SELEX data) to identify potential factors binding to a region of interest and to use the IPOD-HR occupancy maps to obtain initial estimates of the condition-dependent occupancy of any identified sites.

## Global de novo discovery of sequence specificity motifs for active transcription factors

While the peak calls obtained from IPOD-HR data show strong enrichments with known TFBSs (**S1 Fig**), many of the called peaks do not match any known or predicted TFBSs (as detailed in **Fig 5A**) and likely correspond either to unknown sites for well-characterized TFs or binding sites for previously uncharacterized TFs. Given that the majority of the newly inferred binding sites appear not to correspond to known or predicted sites for annotated TFs, we hypothesized that the regulons corresponding to those motifs would likely show enrich-ments for poorly annotated genes, as we expect here to reveal the regulatory logic driving typi-cally understudied pathways. We thus identified likely regulatory targets of each newly called peak, divided them between poorly annotated genes (those with UniProt annotation scores of 1 or 2 out of 5 [34]) and well-annotated genes, and then examined the proportion of poorly annotated targets for occupancy peaks matching RegulonDB binding sites compared with all

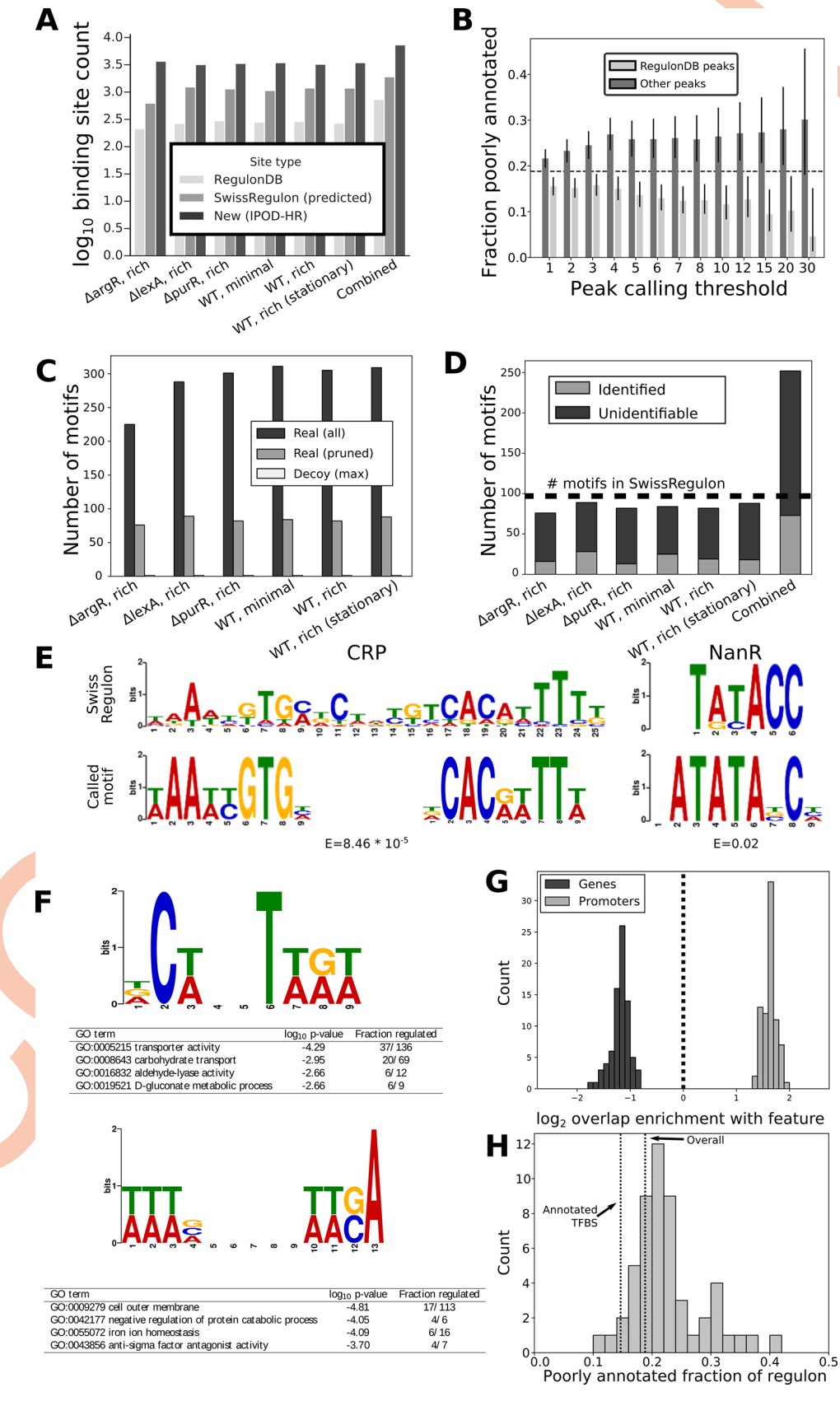

**Fig 5. Genome-wide de novo discovery of sequence specificity motifs for actively bound TFs. (A)** At a peak calling threshold of 4 (cf **S1 Fig**), we show the number of identified binding sites that overlap with annotated sites in RegulonDB ("RegulonDB"), motif-based predicted binding sites ("SwissRegulon"), or novel ("New"). The "Combined" category represents peak sets where the peaks at a given threshold identified across all conditions are merged, prior to comparisons with the RegulonDB and predicted databases. Qualitatively similar results are observed at all tested peak calling thresholds (all peaks are provided in **S2 Data**). **(B)** All called IPOD-HR occupancy peaks across the conditions shown in panel **A** were combined and then partitioned based on whether they overlap with a known or inferred binding site in RegulonDB (RegulonDB peaks) or not (Other peaks). Peaks were then considered to have regulatory potential if they fell within 100 bp of an annotated transcription start site, and the fraction of the genes potentially regulated by each peak category plotted across different peak calling threshold. Error bars show 95% credible intervals calculated assuming that the incidence of poorly annotated genes in the inferred regulon is a binomial random variable, using Bayesian inference with a Beta(1,1) prior. The dashed line shows the overall fraction of poorly annotated genes included in the analysis (i.e., those belonging to transcripts regulated by at least 1 annotated transcription start site in RegulonDB). **(C)** Number of motifs discovered de novo using IPOD-HR occupancies under each condition in our study. "All" and "pruned" refer to all discovered motifs and those surviving cluster-based filtering by RSAT (see Methods for details), respectively. "Real" shows the motif counts discovered in real data, and "Decoy" shows the maximum discovered motif count across 20 independent circular permutations of the data under each condition. **(D)** Classification of nonredundant motifs across conditions as "Identified" (match to an existing motif from the SwissRegulon database, via TOMTOM, with E-value < 0.5) or "Unidentified" (no matches found with E < 0.5). "Combined" refers to the full set of motifs discovered after pooling all motifs across all conditions and redundancy filtering; a horizontal dashed line shows the total number of known motifs present in SwissRegulon. **(E)** Example cases of "Identified" matches of IPOD-HR-inferred motifs with motifs from the SwissRegulon database, showing good correspondence with annotated CRP (left) and NanR motifs. E-values arising from the TOMTOM search pairing newly discovered motifs with similar known motifs are shown beneath each inferred motif. *y* axes for motifs in this and the following panel show information content in bits. In the case of CRP, the half site was inferred and is shown here in both the forward and reverse orientations aligned to the motif in SwissRegulon. **(F)** Examples of 2 newly inferred motifs that do not have identifiable hits in the SwissRegulon database (as assessed using TOMTOM). In each case, representative GO terms showing significant enrichments amid the predicted regulon associated with that motif are shown (see Methods for details). **(G)** Overlap of predicted binding sites for IPOD-HR inferred motifs with either coding regions (genes) or promoters (both as annotated in RegulonDB) using only strict motif hits; shown are the $\log_2$ fold enrichment or depletion of the overlap as compared with that expected by chance. **(H)** For the predicted regulon of each newly inferred motif (using only strict motif hits), we show the fraction of regulon members that are poorly annotated (UniProt annotation score of 1 or 2 out of 5); for comparison, dashed lines are shown for the values obtained when the same statistic is calculated for all annotated TF–gene interactions in RegulonDB ("Annotated TFBS") and for the genome as a whole ("Overall"). GO, gene ontology; IPOD-HR, in vivo protein occupancy display— high resolution; TF, transcription factor; WT, wild-type.

other peaks. As shown in **Fig 5B**, peaks that do not correspond to RegulonDB-annotated binding sites are strongly enriched upstream of poorly annotated genes, whereas those matching annotated binding sites are enriched for well-annotated genes. Thus, examination of occupancy peaks derived from IPOD-HR enables identification of a large number of new putative regulatory sites, with a particular abundance of possible regulators of poorly annotated genes.

Our large-scale identification of new TFBSs also raises the important possibility that new TF binding motifs might likewise be identifiable through de novo computational motif discovery in the set of all sequences within IPOD-HR peaks. Indeed, the application of the FIRE [35] motif discovery algorithm to peak locations obtained from IPOD-HR data reveals hundreds of de novo discovered sequence motifs that are informative of strong occupancy sites, even after pruning of redundant motifs (**Fig 5C**). Upon cross-referencing with a database of known *E. coli* TFBS motifs using TOMTOM [36], we find that approximately 25% of the discovered motifs can be matched with known motifs (notably, 86/97 of the annotated motifs in the *E. coli* SwissRegulon database are matched by at least 1 inferred motif from the set present prior to redundancy pruning and 68/97 match at least 1 motif present in our inferred set after pruning), while, at the same time, nearly 200 novel motifs are called with similar confidence (**Fig 5D**). A comprehensive list of redundancy-pruned motifs called across all conditions in our study is given in **S3 Data**. To provide estimates of the false discovery rate (FDR) arising from our motif inference, we performed an identical motif discovery procedure for each biological condition on 20 "decoy" data sets in which the underlying *E. coli* genomic sequence was

rotated by a random distance relative to the peak calls, thus preserving the correlation structure of both the data and sequence with respect to themselves (light bars in **Fig 5C**). Our decoy data sets gave rise to no more than 1 motif under any condition, giving rise to an effective FDR (across shuffles and conditions) of less than 0.5% even for our fully pruned motifs. Using only the novel motifs (i.e., motifs that did not have detectable similarity to any motifs in the Swiss-Regulon database) in a genome-wide search for potential binding sites using FIMO, we find that 84.8% of all IPOD occupancy peaks at a peak calling threshold of 4 can be explained by binding sites for the novel motifs, compared with 7.5% that can be explained by annotated binding sites from RegulonDB (predicted binding sites for all newly inferred motifs from the IPOD-HR data are enumerated in **S4 Data**). Thus, the newly inferred motifs provide a substantially expanded ability to assign the observed profile of protein binding across the chromosome.

In **Fig 5E**, we show 2 representative examples of discovered motifs that show strong matches with annotated motifs, demonstrating that the motifs for well-characterized transcriptional regulators such as CRP and NanR can be inferred directly from IPOD-HR data. For comparison, in **Fig 5F**, we show 2 newly inferred motifs that do not match any known motifs in the *E. coli* SwissRegulon database. Intriguingly, the pattern of binding sites across the *E. coli* chromosome for both of these novel motifs illustrates a potential regulatory function, with the first motif associating with a substantial fraction of the genes involved in import and metabolism of low preference carbohydrates and the second apparently involved in iron acquisition and regulation of protein catabolism. In order to provide additional insight into the potential physiological roles of the factors binding these motifs, we considered the similarity of their putative regulons to those of well-characterized TFs, making use of the Jaccard index comparing the gene sets potentially regulated by each motif with those of known TFs. For the top motif shown in **Fig 5F**, the best match is for CRP, with a Jaccard index of 0.19. This finding is consistent with the putative role in regulating carbon source utilization, although the regulon of the new motif is clearly distinct from those of any well-characterized factor given that the highest Jaccard index observed for it was 0.19, and the next highest hit after CRP was 0.12. Applying the same analysis to the bottom motif shown in **Fig 5F** shows that the strongest detectable similarity in regulons is to those of H-NS (0.12) and Fur (0.11). These matches indicate potential roles for the new motif in coordinating responses to changes in iron starvation and other stress conditions such as changing temperatures, but again, the regulon of the newly identified motif is clearly quite distinct from that of the characterized TF. In order to facilitate similar analysis of the potential functions of the other newly inferred motifs, we provide in **S5 Data** the Jaccard indices for all significant overlaps between the putative regulons of our newly inferred motifs and those of existing factors as annotated in RegulonDB (see caption for details). In addition, we provide an extended discussion of the overlaps between known TFBSs and predicted binding sites for the new motifs in **S2 Text**.

We further assessed the regulatory capacity of all newly called sequence motifs by comparing their genome-wide distribution of binding sites with annotated genes (coding regions) and promoters. We would expect that binding sites for functional transcriptional regulators would be enriched within promoters and depleted from coding regions, as was the case for overall IPOD-HR occupancy (**Fig 2E**). Indeed, the overlap distributions of binding sites for our newly inferred motifs are uniformly enriched for annotated promoters and depleted for open reading frame (ORFs; **Fig 5G**; $p = 1.0 * 10^{-27}$, Wilcoxon rank sum test), demonstrating that motifs inferred directly from IPOD-HR occupancy data occur primarily in likely regulatory regions. Equivalent results were obtained even after excluding all of the newly inferred motifs with identifiable similarity to SwissRegulon motifs ($p = 2.42 * 10^{-19}$, Wilcoxon rank sum test). To further investigate the global regulatory potential of the newly inferred motifs, we applied

iPAGE [37] to identify cases of significant mutual information between predicted binding sites for a given motif (within 100 bp of an annotated transcription start site in RegulonDB) and the corresponding gene ontology (GO) terms of genes in the potentially regulated operons. We found that out of a total of 1,732 motifs with identifiable binding sites, 1,611 have significant mutual information with genes from at least 1 GO term (notably including 163 out of the 176 nonredundant motifs that had identifiable binding sites by our criteria, and no detectable similarity to previously known motifs). These findings highlight potential pathways that may be regulatory targets of these newly inferred motifs; the resulting predicted regulatory targets are summarized in **S6 Data**. In the future, tracking of the occupancy of these motifs over a broader range of physiological conditions would yield additional insight into the likely stimuli sensed by them and the downstream processes that they may regulate (and might allow merging of some of the newly inferred motifs that in fact represent different sequences bound by the same factor).

Given that the majority of the newly inferred motifs appear not to correspond to annotated TFs, and our findings above regarding the enrichment of poorly annotated genes downstream of orphan binding peaks, we hypothesized that the regulons corresponding to our newly inferred motifs would likely show enrichments for poorly annotated genes, as we expect here to reveal the regulatory logic driving typically understudied pathways. We thus calculated the fractions of the hypothetical regulons of each newly inferred motif that consist of poorly annotated genes (defined as noted above). As shown in **Fig 5H**, we found that the regulons of the newly inferred motifs were significantly enriched for poorly annotated genes when compared with both the annotated *E. coli* transcriptional regulatory network in RegulonDB ($p = 6.26 * 10^{-10}$, Wilcoxon signed rank test) and the overall average rate of poorly annotated genes throughout the chromosome ($p = 3.49 * 10^{-5}$, Wilcoxon signed rank test). Pairwise comparison of the rate of poorly annotated genes between the regulon of each newly inferred motif and all genes that were not members of the corresponding regulon likewise showed a significant enrichment for poorly annotated genes ($p = 9.78 * 10^{-6}$, Wilcoxon signed rank test). Taken together, we see that IPOD-HR enables inference of a large number of sequence motifs, many of which likely correspond to functional, but currently understudied, transcriptional regulators in *E. coli*, providing a substantial resource for ongoing investigation of this transcriptional regulatory network.

## Extended protein occupancy domains define distinct and largely stable transcriptionally silent regions with unique sequence features

One of the most striking findings enabled by the original application of IPOD was the discovery of EPODs: large regions of the *E. coli* chromosome that show unusually dense levels of protein occupancy over kilobase or longer scales [10]. EPODs are also clearly apparent in all our IPOD-HR data sets and appear to correspond functionally to the transcriptionally silent extended protein occupancy domains (tsEPODs) of Vora and colleagues [10]. The profile of protein occupancy and EPODs, along with the accompanying impacts on transcript levels, for a representative region of the genome is shown in **Fig 6A**. Indeed, we found that many highly protein occupied regions measured using the original IPOD method (in particular, the highly expressed extended protein occupancy domains or heEPODs) represent RNA polymerase occupancy, whereas the EPODs now revealed by IPOD-HR consist solely of large domains of occupancy by proteins other than RNA polymerase (which typically also exclude RNA polymerase). We discuss these differences and the details of the approach used in IPOD-HR to remove contributions from RNA polymerase in **S1 Text**. The specific resolution of tsEPODs afforded by the IPOD-HR method and the coverage of multiple genetic and nutrient

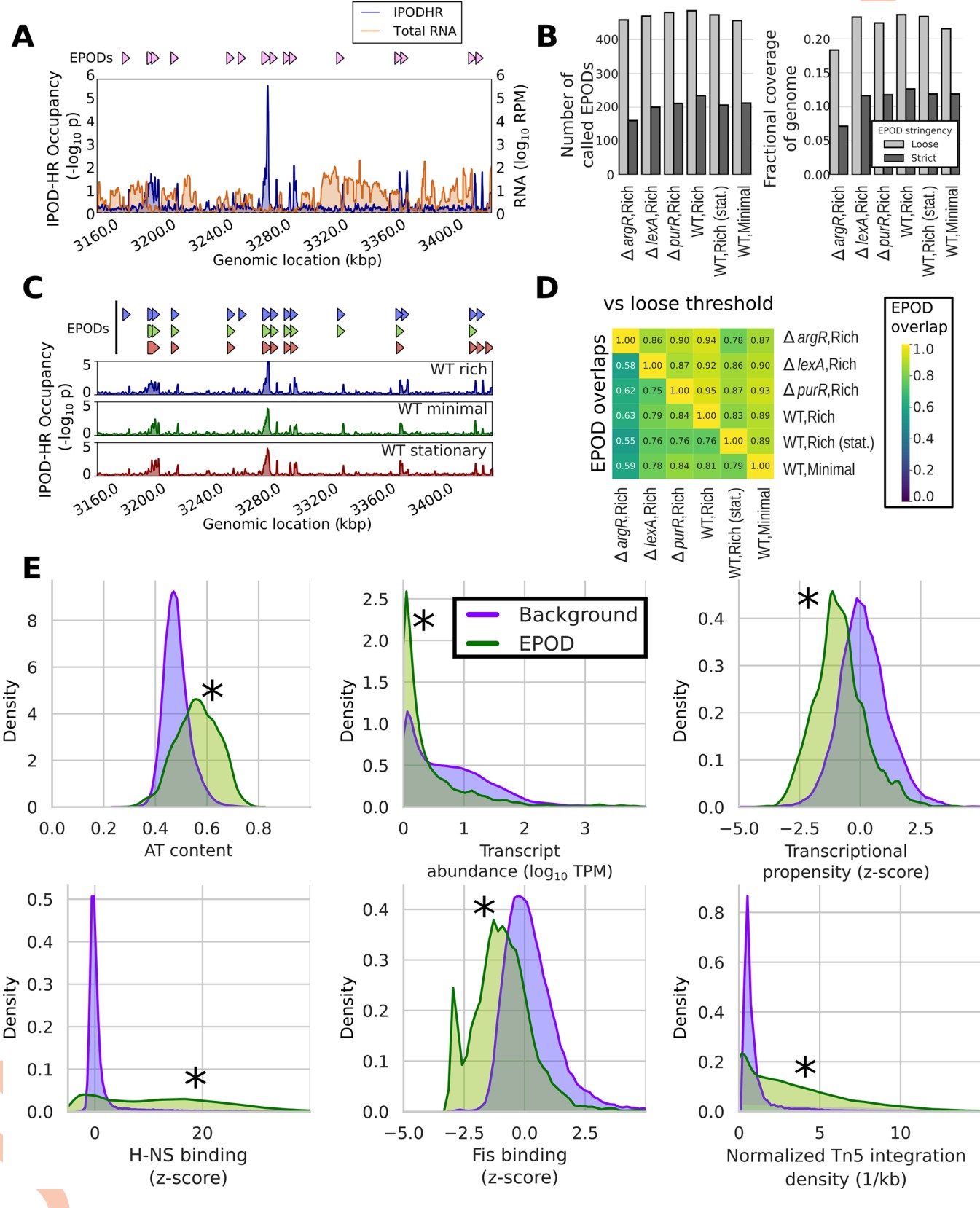

**Fig 6. EPODs define stable genomic structures and are associated with many distinct features. (A)** EPOD calls from a representative genomic region in the WT rich media condition, along with protein occupancy and RNA levels smoothed with a 1-kb rolling median. All displayed/analyzed EPOD calls refer to our strict threshold unless otherwise noted. **(B)** Number of called EPODs by condition (left) and fraction of the genome covered by EPODs (right) for both our loose and strict thresholds (see text for details). **(C)** IPOD-HR occupancies (shown over a 1-kb rolling median) and associated EPOD calls under 3 different conditions, in the same genomic region shown in panel **A**. EPOD calls are shown above the occupancy, in the same order as the data tracks. **(D)** Lower triangle: Overlap of EPOD calls (using a symmetrized distance that is the average of the fraction of EPOD positions from a condition *a* that is also called in condition *b* and vice versa) between each pair (*a*,*b*) of the studied conditions. Upper triangle: Each entry shows the fraction of the EPOD calls (at a 5-bp resolution) from the sample defining that row that is contained in a relaxed set of EPOD calls (see text) of the sample defining that column (only the upper triangle of that matrix is shown; the lower triangle is similar except that the smaller *ΔargR* EPOD set contains fewer of the EPODs from other conditions). **(E)** Density plots showing normalized histograms (smoothed by a kernel density estimator) of the specified quantities for regions of the genome that are in EPODs vs. those that are not (Background), as assessed in the WT M9/RDM/glu (WT,rich) condition. "*" indicates FDR-corrected $p < 0.005$ via a permutation test (against a null hypothesis of no difference in medians). Significance calling and additional comparisons are shown in **S4 Table**. EPOD, extended protein occupancy domain; IPOD-HR, in vivo protein occupancy display—high resolution; WT, wild-type.

perturbations in the present data sets allow us to fully investigate the nature and condition-dependent occupancy of these chromosomal structures.

The identified EPODs show remarkable stability (**Fig 6B**), with approximately 200 EPODs in each physiological condition (or approximately 450 using a relaxed calling threshold; see Methods for details) and similar fractions of the genome contained in EPODs in each case. A comprehensive listing of EPODs identified across our conditions is given in **S7 Data**. The *ΔargR* strain serves as an outlier among the genetic perturbations, with modestly decreased (but still substantial) EPOD coverage, possibly due to decreased expression of H-NS in this condition (RNA polymerase occupancy of the *hns* promoter measured in our ChIP data set is 2-fold lower than in WT cells under equivalent conditions; it is unclear whether this reflects a regulatory or metabolic effect caused by loss of ArgR). The differing behavior of the *ΔargR* strain appears milder using the relaxed EPOD calling threshold, and thus the difference is at least partially just a thresholding effect (**Fig 6B**). The locations of individual EPODs are likewise well maintained, even across very different physiological conditions. For example, in **Fig 6C**, we show IPOD-HR occupancy across the same region as shown in **Fig 6A**, comparing exponential growth in rich versus minimal media and stationary phase cells. In contrast with the condition-dependent occupancy of individual TFs, at the approximately kilobase scale, the occupancy traces are nearly superimposable and show that most EPODs called under the various conditions overlap. Furthermore, out of the subset of EPOD calls that are present in the "WT,rich" condition but absent in the others, all are present in the "WT,minimal" condition using the relaxed EPOD calling threshold, and all but two also present in the stationary phase condition using that threshold, suggesting that many of the small differences in EPOD locations that do appear between EPOD calls under different physiological conditions are in fact due to thresholding effects. We observe the same trends genome-wide: excluding the *ΔargR* case, 71% to 86% of genomic locations (at the base pair level) that are called as EPODs under any one condition are likewise EPOD calls under any other condition (**Fig 6D**), and at least 78% (and typically much more) of the EPODs called in one condition are contained within the relaxed threshold calls under any other condition (*n.b.* the "relaxed" threshold used here corresponds to the original EPOD definition from [10]). It is also worth noting, in this context, that 89% of the tsEPOD-occupied locations from [10] are contained in the new "WT,rich" relaxed threshold EPOD set, in line with the observed concordance across experimental conditions in our new data sets.

Several defining characteristics of EPODs are readily apparent upon cross-referencing with other genome-wide data sets (**Fig 6E**): They represent regions of high AT content, which are both associated with low levels of native transcripts and decreased transcriptional propensity (i.e., expression of standardized integrated reporters [38]). Consistent with our original findings [10], EPODs also show high occupancy of H-NS, HU, and LRP; low occupancy of Fis; and

are associated with high efficiency of Tn5 integration (**Fig 6E**). While the latter might seem surprising given that highly protein occupied regions on eukaryotic chromatin tend to exclude Tn5 (as is used to great effect in assay for transposase-accessible chromatin using sequencing [ATAC-seq] [39]), we note that bacterial H-NS occupancy has previously been shown to facilitate Tn5 activity [40]. Additional characteristics of EPODs, such as reduced densities of possible Dam methylation sites (consistent with the expected blocking of Dam methylase by bound proteins, previously shown in in vivo methylase protection experiments [41]) and a characteristic pattern of DNA structural parameters including decreased minor groove width, are shown in **S4 Table**.

The remarkable condition invariance of the locations of EPODs outlined above, even across such dramatic changes as transition from exponential to stationary phase, suggests that EPODs predominantly represent fixed structural features of the *E. coli* chromosome, rather than highly dynamic regulatory structures. We thus examined the classes of genes (assessed using GO terms) most strongly enriched or depleted in EPODs. As illustrated in **Fig 7A**, EPODs show strong enrichments for mobile elements (GO:0006313) and prophage genes (specifically lytic pathways; GO:0019835) and are depleted for core metabolic pathways such as ribosome components (GO:0005840). Indeed, EPODs are associated with the silencing of many prophages (e.g., **Fig 7B**) and even smaller operons of unknown function (e.g., **Fig 7C**).

Our findings regarding EPODs, particularly the high levels of H-NS binding in EPODs and the known role of H-NS as a xenogeneic silencer [42,43], are highly consistent with prior information regarding the silencing role of H-NS. In order to determine the extent to which H-NS silenced regions and EPODs as defined here overlap, we compared the distribution of EPODs across the genome with H-NS ChIP-seq data from [44]. Using an unsupervised clustering method to divide genomic intervals into high, medium, and low levels of H-NS occupancy, we found that 72.4% of EPODs fall into the high H-NS category, compared with 5.4% of non-EPOD regions (**S3 Fig**, panel A). Nevertheless, when considering the average transcript levels observed as arising from the same genomic intervals, the EPODs from the low H-NS and medium H-NS categories still showed significantly lower expression than non-EPOD regions with similar H-NS levels (**S3 Fig**, panel B), and the small number of highly H-NS bound regions that are not part of EPODs are in fact more silent than highly H-NS bound EPODs. Taken together, we thus observe that while many EPODs represent chromosomal regions silenced by H-NS, roughly one-third to one-fourth of EPODs do not show the characteristics of highly H-NS occupied regions, but are nevertheless transcriptionally silenced by an extended stretch of high protein occupancy. The possibility of course exists that H-NS is repositioned in the conditions of our study, which differ from those of [44], to cover the remainder of the EPODs identified here. The mechanism of silencing at these non-H-NS dependent EPODs will likely be a fruitful area for future investigation.

## Discussion

The study of bacterial transcriptional regulatory networks has long benefitted from bottom-up approaches such as DNase footprinting, ChIP-chip, and ChIP-seq to map the behavior of individual factors and regulons. At the same time, however, the insight provided by such approaches has been inherently limited by the need to specify a priori the target of investigation, either in terms of the regulator, regulated gene, or both. However, as we hope to have demonstrated here, a global agnostic strategy (as exemplified by IPOD-HR) provides a unique top-down complement to existing methods by permitting rapid profiling of the protein occupancy landscape of a bacterial chromosome. We have demonstrated that IPOD-HR simultaneously enables resolution of individual changes in TF binding at specific sites, inference of

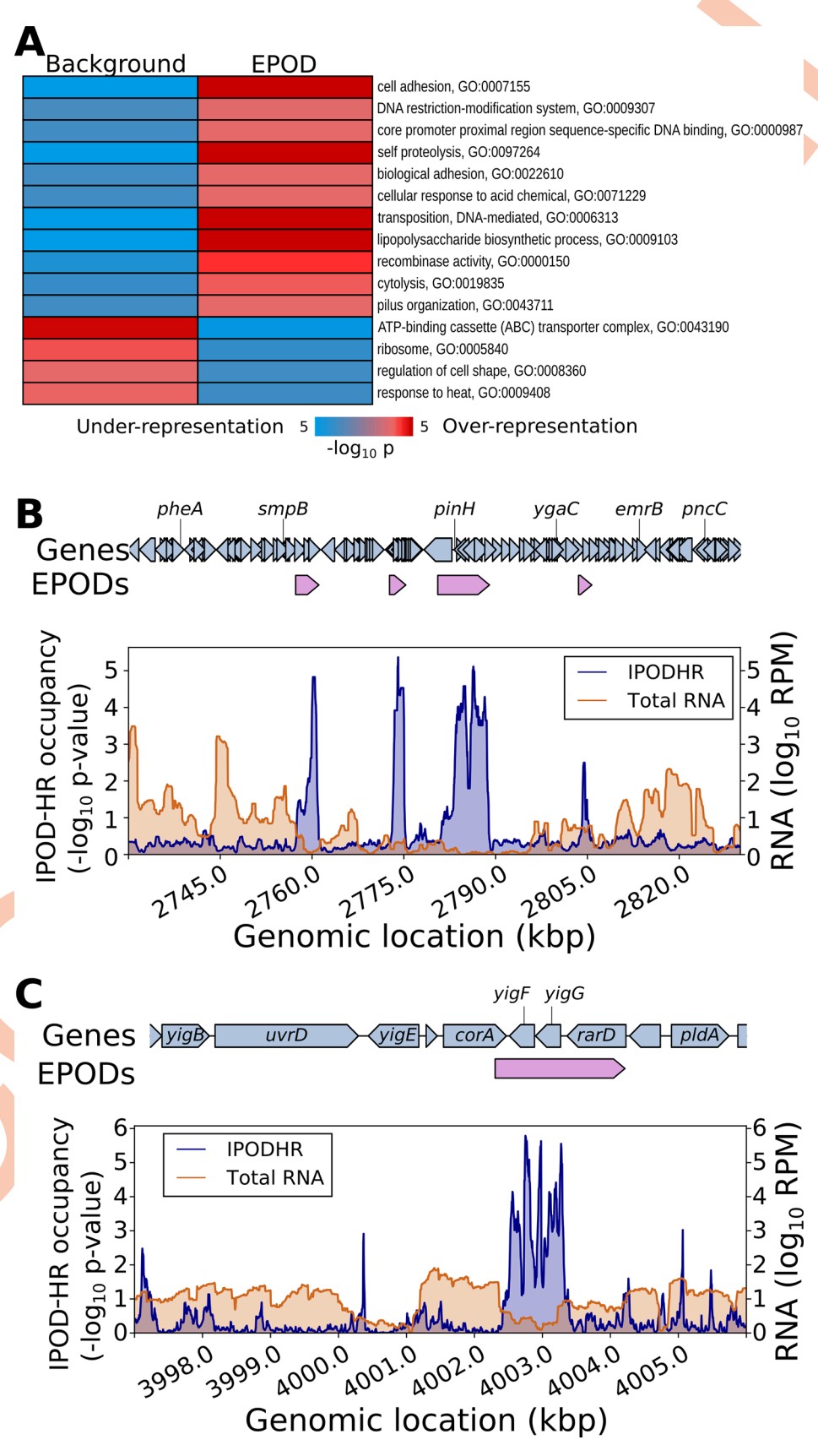

**Fig 7. EPODs are statistically enriched for genes in specific functional categories. (A)** The genome was split into EPOD and background regions as in **Fig 6**; we then applied iPAGE [37] to identify GO terms showing significant mutual information with occupancy in EPODs. All shown GO terms were significant according to the built-in tests in iPAGE. **(B)** Multiple EPODs are associated with silencing of the CP4-57 prophage. Shown are the IPOD-HR occupancy and transcript levels in the vicinity of the prophage locus during growth in rich defined media with glucose, with EPOD locations indicated above the plots. **(C)** Association of a small EPOD with 2 genes of unknown function, *yigF* and *yigG*, along with the putative transporter *rarD/yigH*; data tracks defined as in panel B. EPOD, extended protein occupancy domain; GO, gene ontology; IPOD-HR, in vivo protein occupancy display—high resolution.

new regulatory motifs that likely correspond to functional but poorly characterized transcriptional regulators, and large-scale patterns of protein occupancy indicative of constitutively silenced genomic regions. IPOD-HR thus falls into the same family as methods such as DNase I hypersensitivity [45], micrococcal nuclease digestion with deep sequencing (MNase-seq) [46], and ATAC-seq [39], but was from its inception developed, tuned, and validated for the unique molecular and biophysical features of bacterial chromosomes (we note that applications of ATAC-like methods to bacteria have recently been reported in preprints [47,48] postdating the original preprint of the present manuscript [49]).

We expect that all 3 key capabilities of IPOD-HR highlighted above will prove to be of substantial utility in investigating all cultivable bacterial transcriptional regulatory networks and could potentially even be applied to environmental samples to study occupancy landscapes in uncultivable bacteria. The ability to directly track the occupancy of TFBS for a large set of transcriptional regulators in parallel provides the missing link that has previously stymied efforts to predict the transcriptional output of *E. coli* across conditions, as consideration of only the expression levels of TFs to predict the behavior of their regulons has yielded mixed results [8,50]. Furthermore, the ability to identify likely regulatory sites even in the absence of prior knowledge, as shown both for isolated promoters (**Fig 4**) and inference of entire regulons (**Fig 5**), will substantially accelerate our ability to complete a wiring diagram for the *E. coli* transcriptional regulatory network and to rapidly approach the networks of other less well-characterized bacterial species. In the latter case, we expect that the ability of IPOD-HR to highlight the locations of TFBSs, and the demonstrated feasibility of inferring DNA-binding protein sequence motifs based on those sites, will allow the rapid identification of sites of interest for detailed follow-up studies (e.g., application of pulldowns similar to those in **Fig 4** to identify the precise factor(s) occupying a particular motif). The resulting information will provide an essential set of building blocks to aid in the overall reconstruction of a transcriptional regulatory network. In addition, tracking of the occupancy of different motifs across conditions will likely assist in the identification of biological conditions pertinent to the function of the factor binding each motif, and cross-referencing with RNA polymerase occupancy will provide evidence on the sign of regulatory output associated with each site. IPOD-HR thus provides a powerful high-throughput in vivo approach tracking occupancy at native sites, complementing methods based on screening with purified proteins [29], computational inference [31], or reporter assays [30].

Our study of diverse experimental conditions across different genetic and physiological states provides a comprehensive view of the protein–DNA interactome of *E. coli*. As we have shown, the majority of discovered occupancy events do not correspond to previously known or annotated sites of protein–DNA interactions. We have further shown that these global occupancy profiles can be used for wholesale discovery of sequence specificity for the set of TFs active under these conditions. These occupancy maps and the corresponding DNA motifs provide the community with a rich catalog of likely regulatory events to study, targeting either particular genes or larger pathways. Indeed, our finding that the novel occupancy sites and

DNA motifs are highly enriched upstream of genes that are understudied promises to discover and expand the physiological and regulatory modules of *E. coli*.

Our study also provides significant additional evidence for the presence of large, transcriptionally silent, high-occupancy chromosomal domains in *E. coli*, termed "EPODs." Many such EPODs clearly correspond to regions of H-NS binding, which has previously been shown to form several types of filaments that silence horizontally acquired DNA [5,43,51–53]. On the other hand, we also observe a substantial fraction of EPODs that do not correspond to H-NS binding and yet are still associated with transcriptionally silent regions of the chromosome. Numerous questions regarding the nature and role of those EPODs remain for future work, including the following: What is the protein composition of non-H-NS EPODs? What rules dictate their formation on specific sites? We are also tempted to speculate that, in some contexts, non-H-NS EPODs may undergo condition-dependent changes in occupancy that drive transcriptional regulation, although no such cases could be definitively identified in the conditions studied here. Such behavior has already been observed for H-NS filaments in various enterobacteria [54,55]. Ongoing application of IPOD-HR to a broader range of physiological conditions in *E. coli* will provide further insight into the overall landscape of large-scale protein occupancy across conditions, allowing tracking both of occupancy associated with H-NS (and the related protein StpA) and other classes of EPODs in a single experiment.

Our IPOD-HR strategy for mapping the global dynamics of the *E. coli* protein–DNA interactome relies only on simple physicochemical principles for isolating protein–DNA complexes. As such, it is easily transferable to other bacterial species. The rich and comprehensive data sets that would result, and application of statistical inference during data processing as exemplified here, will provide important regulatory roadmaps in organisms with less well-studied transcriptional regulatory networks, such as non-model bacterial species of clinical and industrial importance. In the future, applications to a broader range of physiological conditions (in *E. coli*) and to other bacterial strains and species will provide important information on the role of large-scale nucleoprotein assemblies on gene regulation and pave the way for more comprehensive and predictive models of transcriptional regulatory logic.

## Methods

### Strain construction

The base strain for all experiments used here is an MG1655 stock obtained from the Tavazoie Lab, which belongs to the substrain typified by ATCC 700926 [56]. All specified gene knockouts were obtained by P1 transduction [57] of the FRT-flanked *kanR* marker from the corresponding knockout strain of the Keio collection [58], followed by Flp recombinase mediated excision of the marker using the pCP20 plasmid [59] to leave a small scar in place of the original ORF. Candidate isolates for each deletion were grown overnight at 42˚C to drop the pCP20 plasmid and then replica plated onto appropriate selective plates to ensure loss of both the plasmid and kanamycin resistance marker. Knockouts were confirmed by PCR fragment sizing and/or sequencing across the marker scar. Note that the Δ*lexA* strain that we refer to is in fact Δ*lexA*/Δ*sulA*, as loss of *lexA* is lethal in the presence of a functional *sulA* gene [60,61].

For construction of the *lacZ* reporter strains for the experiments shown in **Fig 4E**, the *lacZ* gene from our MG1655 background strain was first removed by P1 transduction of a chloramphenicol resistance gene, yielding MG1655 *lacZ::cml*. An additional copy of this strain was constructed by transduction of the Keio collection *yieP::kan* gene and subsequent marker excision (as above) to yield MG1655 *lacZ::cml yieP::scar*. We then generated plasmid-born copies of a reporter gene in which the *lacZ* ORF was placed directly after the *sdaC* promoter of *E. coli* MG1655 (positions 2927672–2928228 on the U00096_3 reference genome). We also generated

promoter variants in which either or both of the identified putative YieP binding sites in the promoter were perturbed, with Site 1 "catttcAT**TTG**TTATATG**AAT**gtttctt" to "catttcAT**AAC**TTATATG**TTA**gtttctt" containing a mutation and Site 2 containing a "cagttaAT**ATG**TCATAC**AATT**tatgttg" to "cagttaAT**GAC**TCATAC**TTCA**tatgttg" mutation (in each case the capitalized region is the putative YieP site; Site 2 corresponds to site highlighted in cyan in **Fig 4D**, with Site 1 the site further upstream of it relative to *sdaC*). All 4 promoter variants were generated by Gibson assembly [62] using the NEB Hifi Builder Master Mix on a plasmid next to an FRT-flanked kanamycin resistance marker, cloned into DH5α cells, and validated by Sanger sequencing of the promoter and adjacent plasmid regions. The promoter-*lacZ* constructs were then each integrated into a WT MG1655 strain replacing the *araC-araBAD* locus via λ_red recombination of a dsDNA PCR product produced using NEB Q5 polymerase (using pKD46 as a helper plasmid as above), selected on MacConkey/arabinose + kanamycin plates, and transduced into the MG1655 *lacZ::cml* and MG1655 *lacZ::cml yieP::scar* strains. The kanamycin resistance marker was removed following the standard pCP20-based marker excision protocol as above, and the final 8 strains (all pairwise combinations of 4 promoter variants and *yieP*+/−) were validated by PCR fragment sizing of the *yieP* and *ara* loci and Sanger sequencing of the *sdaC* promoter at the *ara::sdaCp-lacZ* locus.

## Media/culture conditions

For routine cloning applications and for recovery of cryogenically preserved cells, we used LB (Lennox) media (10 g/L tryptone, 5 g/L yeast extract, and 5 g/L NaCl), with bacteriological agar (15 g/L) added as appropriate.

For physiological experiments, we made use of a variety of supplemented versions of M9 defined medium (6 g/L $Na_2HPO_4$, 3 g/L $KH_2PO_4$, 1 g/L $NH_4Cl$, 0.5 g/L NaCl, 1 mM $MgSO_4$) [56]. Our M9 minimal media condition (M9/min) additionally includes 0.2% (w/v) glucose, 0.4 mM $CaCl_2$, 40 μM ferric citrate, and the micronutrient mixture typically incorporated in MOPS minimal media [63]. Our M9 RDM condition (M9/rdm) instead incorporates into the M9 base 0.4% (w/v) glucose, MOPS micronutrients (as above), 4 μM $CaCl_2$, 40 μM ferric citrate, and 1x supplements ACGU and EZ as used in MOPS RDM [63].

## Cell growth and harvest for IPOD-HR

The cells of interest were grown overnight in the media of interest after inoculation from an LB plate. In the morning, the culture was back-diluted into fresh, prewarmed media to an OD600 of 0.003. The culture was then grown to the target OD600 (0.2, except in the case of stationary phase samples, which are described below), at which point a 200-μL aliquot was removed and preserved in 1 mL of DNA/RNA Shield (Zymo Research) following the manufacturer's instructions.

The remainder of the culture was treated with rifampin to a final concentration of 150 μg/mL and incubated for 10 minutes under the same culture conditions as the main growth to immobilize initiating RNA polymerase at active promoters and permit completion of transcripts in progress. The culture was then rapidly mixed with concentrated formaldehyde/sodium phosphate (pH 7.4) buffer sufficient to yield a final concentration of 10 mM $NaPO_4$ and 1% v/v formaldehyde. Cross-linking was allowed to proceed for 5 minutes at room temperature with vigorous shaking, followed by quenching with an excess of glycine (final concentration 0.333 M) for 5 minutes with shaking at room temperature. The cross-linked cells were subsequently chilled on ice and washed twice with ice-cold phosphate buffered saline, 10 mL per wash. The fully washed pellets were carefully dried, any remaining media pipetted away, and then the pellets were snap-frozen in a dry ice-ethanol bath and stored at -80 C.

In the case of our stationary phase samples, cells were grown as described above in terms of back-dilution and growth to an OD600 of 0.2, and then grown for an additional 3 hours prior to RNA harvest, rifampin treatment, and cross-linked as described above.

## Cell lysis and DNA preparation

Frozen cell pellets were resuspended in 1x IPOD lysis buffer (10 mM Tris HCl, pH 8.0; 50 mM NaCl) containing 1x protease inhibitors (Roche Complete Mini, EDTA free) and 52.5 kU/mL of ready-lyse (Epicentre): 600 μL per pellet (stationary phase cells were diluted 10× prior to lysis, and only 1/10 of the resulting material used, due to the much higher biomass of those pellets). We incubated the resuspended pellet for 15 minutes at 30°C and then placed it on ice. We then sonicated the cells using a Branson digital sonicator at 25% power, using three 10-second bursts with 10-second pauses between bursts. The cells were maintained in a wet ice bath throughout sonication.

We then performed a calibrated DNA digestion to sub-200-bp fragments, by adding to the sonicated lysates 60 μg RNase A (Thermo Fisher Scientific), 6 μL DNase I (Fisher product #89835), 5.4 μL 100 mM MnCl2, and 4.5 μL 100 mM CaCl2, and then incubating on ice. While the appropriate digestion time must be calibrated for each particular sample type and batch of DNase, 30 minutes of digestion proved appropriate for all samples here. Reactions were quenched after completion by the addition of 50 μL 500 mM EDTA (pH 8.0), typically yielding 50- to 200-bp fragments.

## IPOD-HR interface extraction

Prior to interface extraction, samples were clarified by centrifugation for 10 minutes at 169,000 x g at 4 C. After clarification, a 50 microliter input sample was diluted 1:9 in elution buffer (50 mM Tris, pH 8.0; 10 mM EDTA; 1% SDS) and kept on ice until the reverse cross-linking step. The remainder of the lysate was mixed with 1 volume of 100 mM Tris base and 2 volumes of 25:24:1 phenol:chloroform:isoamyl alcohol, vortexed, and then incubated for 10 minutes at room temperature. After incubation, the sample was spun at 21,130 x g for 2 minutes at room temperature, allowing formation of a white disc at the aqueous–organic interface enriched for protein–DNA complexes [10,11].

The complete aqueous phases were removed and discarded, and the remaining disc washed again with 350 microliters TE (10 mM Tris, pH 8.0; 1 mM EDTA), 350 microliters 100 mM Tris base, and 700 microliters 24:1 chloroform:isoamyl alcohol. The resulting mixture was vortexed vigorously and again centrifuged for 2 minutes at 21,130 x g. All liquid was again removed, and the wash was repeated using 700 microliters TE and 700 microliters 24:1 chloroform:isoamyl alcohol. After vortexing, centrifugation, and removal of the final wash (exactly as above), any residual liquid was removed by wicking with a laboratory wipe (if any substantial pools of liquid were present). Finally, the interface was resuspended in 500 microliters of elution buffer (described above), vortexed vigorously, and kept on ice until reverse cross-linking (no more than a few hours).

We caution the reader that the separation of the interface layer from the liquid on either side of it is crucial to success with this method. We have found it most effective to tilt the microcentrifuge tube toward while pipetting out the organic layer from beneath, at which point the interface will adhere to the tube wall and allow easy removal of the aqueous layer. We have also found that the handling characteristics of the interface vary greatly with the plasticware in use. For the work described here, we have used 2-mL microcentrifuge tubes from USA Scientific for all interface handling, as the interfaces adhere nicely to the tube wall (other plasticware may yield variable results); at the same time, the use of low-retention pipette tips appears to reduce binding of the interface to the tip.

## RNA polymerase chromatin immunoprecipitation

DNA for RNA polymerase ChIP-seq experiments was prepared as described above for IPOD-HR interface extraction up through the lysate clarification stage. Whenever possible, we used frozen pellets obtained from the same culture for matched IPOD-HR and ChIP-seq experiments, in which case the lysates were pooled and mixed immediately prior to removal of a single input sample. ChIP procedures here were modeled on those of [64].

The digested lysates were mixed 1:1 with 2x IP buffer (200 mM Tris, pH 8.0; 600 mM NaCl; 4% Triton X-100; 2x Roche Complete EDTA-free protease inhibitors) and then kept on ice for no more than a few hours prior to antibody addition. We added 10 microliters of purified anti-*E. coli* RNA polymerase antibody (NeoClone WP023) and incubated overnight with rocking at 4˚C. Near the end of the incubation period, we resuspended an aliquot of 50 microliters of protein G dyna-beads (Invitrogen) and equilibrated the protein G beads with 1x IP buffer lacking protease inhibitors. The bead aliquot was added to the antibody-lysate mixture and then incubated 2 hours with rocking at 4˚C. The bead–antibody–target complexes were subsequently subjected to the following series of washes, with 1 mL used per wash. All washes were at room temperature and involved manual resuspension of the beads in the new wash buffer followed by immediate re-separation:

- 1x Wash buffer A (100 mM Tris, pH 8.0; 250 mM LiCl; 2% Triton X-100; 1 mM EDTA)

- 1x Wash buffer B (100 mM Tris, pH 8.0; 500 mM NaCl; 1% Triton X-100; 0.1% sodium deoxycholate; 1 mM EDTA)

- 1x Wash buffer C (10 mM Tris, pH 8.0; 500 mM NaCl; 1% Triton X-100; 1 mM EDTA)

- 1x TE (10 mM Tris, pH 8.0; 1 mM EDTA).

The antigens were subsequently eluted by adding 500 microliters of elution buffer (composition described above) and incubating 30 minutes at 65˚C, with vigorous vortexing every 5 to 10 minutes.

## Cross-linking reversal and recovery of DNA

The DNA from the input, IPOD-HR, and ChIP fractions described above was recovered using identical procedures: Samples diluted in elution buffer (see above) were incubated overnight (6 to 16 hours) at 65˚C to reverse formaldehyde cross-links. After allowing the samples to cool to room temperature, we then added 100 μg of RNase A (Thermo Fisher Scientific), incubated 2 hours at 37˚C, then added 200 μg of proteinase K (Fermentas) and incubated an additional 2 hours at 50˚C. DNA was then recovered via standard phenol–chloroform extraction and ethanol precipitation, following protocols from [57]. We used Glycoblue (Ambion) as a coprecipitant, NaCl as a precipitating salt (due to the presence of SDS in our solution), and washed with ice-cold 95% ethanol to avoid loss of low molecular weight DNA.

Recovered DNA was quantified via fluorescent quantitation (using either the Invitrogen PicoGreen or Promega QuantIT system), and samples of sufficiently high concentration were also run on a 2% agarose gel for fragment size assessment. Typical total yields from the procedure above were on the order of 1 μg of DNA for the input samples, 100 to 200 ng for the IPOD-HR samples, and 1 to 10 ng for the ChIP samples.

## Preparation of next-generation sequencing (NGS) libraries

Except as otherwise noted, all DNA samples were prepared for Illumina sequencing using the NEBNext Ultra DNA Library Prep Kit (NEB product #E7370), with either single index or dual index primers also obtained from NEB. We followed the manufacturer's instructions except for the following variations:

- Cleanups prior to the PCR step were performed using a Zymo Clean&Concentrator 5 spin column kit or Zymo Oligo Clean&Concentrator spin column kit instead of Ampure beads, in order to avoid the loss of low molecular weight DNA.

- We used Ampure and Axygen PCR cleanup beads interchangeably. The final cleanup step was in some cases repeated to remove obvious populations of adapter dimers.

All libraries were sequenced on either an Illumina HiSeq or NextSeq instrument; detailed statistics on read lengths and counts are provided in **S5 Table**. A small number of samples were prepared for sequencing using an Illumina Truseq Nano kit or (in the case of one biological replicate) a Truseq kit instead of the NEBnext kit noted above (those samples are identified in **S5 Table**); we found that upon calculation of correlations between the coverages of a broad range of IPOD, input, and RNA polymerase, ChIP-seq samples prepared using various sequencing preparation kits that the Truseq Nano samples were indistinguishable from NEB-Next Ultra samples upon clustering of the occupancy profiles across sample types and conditions. To avoid any possibility of bias from the single included Truseq replicate (which was for the Δ*lexA* genotype), we observed qualitatively equivalent results in TFBS occupancy dynamics (**Fig 3A**) if the Truseq replicate was removed from analysis (notably, the rank ordering of LexA occupancy across samples was preserved, and of the 6 conditions considered, only the stationary phase and Δ*lexA* conditions had average occupancy $-\log_{10}p$ scores less than 2).

## RNA isolation and RNA-seq sample preparation

As noted above, samples for RNA isolation were preserved immediately prior to rifampin addition by dilution in a 5x excess of DNA/RNA Shield (Zymo Research); the RNA samples were then stored at −80˚C until purification. RNA was isolated using a Zymo QuickRNA microprep kit following the manufacturer's instructions, including the on-column DNase digestion. Purified RNA was quantified using RiboGreen (Invitrogen) and then ribosome-depleted using the Illumina RiboZero Gram-negative bacteria kit according to the manufacturer's instructions, with the input RNA amount and all reaction volumes cut in half. Final recovery of the ribo-depleted RNA was accomplished using the modified Zymo spin column protocol present in the RiboZero documentation. Ribo-depleted RNA was then prepared for sequencing using the NEBNext Ultra Directional RNA kit (NEB product E7420) and sequenced as described above for the DNA samples.

## Analysis of NGS data

All NGS data were preprocessed using a common pipeline, after which DNA and RNA data sets were processed separately. The reference genome in all cases was the most recent version of the *E. coli* MG1655 genome (GenBank U00096.3), with gene, TFBS, and transcription start site annotations from RegulonDB [7]. Data processing was automated using in-house python and bash scripts and parallelized where possible using GNU parallel [65] or the python multiprocessing library. The python and bash source code used to derive processed occupancy data from raw reads is available from https://github.com/freddolino-lab/ipod_v1_2020; we also have distributed a ready-to-use analysis environment in the form of a singularity container, as documented in the GitHub repository noted above.

**Read quality control and preprocessing.**   All reads were subjected to adapter removal using cutadapt 1.8.1 [66] to cut the common sequence of Illumina Truseq adapters and then trimmed to remove low-quality read ends with Trimmomatic 0.33 [67], using the trimming steps "TRAILING:3 SLIDINGWINDOW:4:15 MINLEN:10." Samples were subjected to

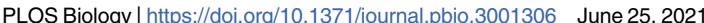

additional manual quality checks using FastQC [68] and MultiQC [69] to identify any irregularities in terms of sequence content, quality, or duplication.

**DNA sequencing and protein occupancy calling.** Surviving DNA reads were aligned to the U00096.3 genome using bowtie2 version 2.1.0 with "very sensitive" end-to-end alignment presets, and dovetail alignments allowed. Only concordant paired-end reads were retained for subsequent quantitation. Read occupancies were calculated at a 5-bp resolution along the chromosome using the parsing method of Kroner and colleagues [24] and scaling the base pair–wise contribution of each read by the inverse of its length (thus, each read contributed the same total amount of occupancy signal). The resulting read densities were then quantile normalized, acting separately for the input, IPOD interface, and RNA polymerase ChIP-seq tracks of each biological condition. In order to account for the higher amount of genomic DNA present near the origin of replication relative to the terminus, we fitted a smoothing spline with 4 evenly spaced knots and periodic boundary conditions to the input sample for each condition; the smoothing spline provides a low-pass filter on abundances that accounts for large-scale variations in DNA abundance across the genome. The use of 4 knots allows inflection points at the origin, terminus, and the halfway points between them. All occupancy data sets were divided by the spline-smoothed abundances of the corresponding input data prior to further processing. After abundance normalization, all data tracks were rescaled to have matching means, and then all replicates for each sample type/biological condition combination were averaged to generate a composite occupancy track (yielding, for example, 1 input data track for the WT M9/RDM/glu condition, 1 IPOD data track for the WT M9/RDM/glu condition, etc.).

The displayed IPOD and ChIP data tracks were then obtained as $\log_2$ ratios of the extracted (interphase or ChIP) to input samples for each condition; we refer to these tracks as the "IPOD" and "ChIP" signals below. Upon viewing the correlation between total protein occupancy and RNA polymerase occupancy, 2 protein-occupied subpopulations were apparent (**S4 Fig**): a linear subpopulation where total protein and RNA polymerase are well correlated and a second subpopulation of positions where the total protein occupancy is much higher than expected based on the RNA polymerase occupancy. We interpret the former set of positions as protein occupancy due directly to RNA polymerase binding and the latter as non-RNA polymerase occupancy (as schematized in **Fig 1B**). To obtain the fully processed IPOD-HR signal for non-RNA polymerase occupancy, we applied a modified linear model to estimate and then remove the contributions of RNA polymerase to the observed signal. To this end, we began by using only the top 2% of observed ChIP values, which are presumed to represent sites that are bound entirely by RNA polymerase in the IPOD versus ChIP signal comparison (as in **S4 Fig**). We then generated the line with the lowest possible slope, and zero intercept, that is sufficient to keep 95% of the high-ChIP data region below it; thus, this represents a linear transformation of the ChIP signal that is sufficient to remove the vast majority of RNA polymerase occupancy. The resulting slope of the IPOD versus ChIP data was then used to generate a prediction of the IPOD contribution attributable to RNA polymerase binding at each position based on the observed ChIP signal itself and that value subtracted from the IPOD signal. No subtraction was performed at positions where the observed RNA polymerase occupancy was negative, so that the ChIP subtraction can only lower (not raise) occupancy. We note that the choice of slope to completely eliminate all occupancy at 95% of the high-ChIP sites represents a highly conservative choice (in terms of removing virtually all occupancy from the IPOD signal that could potentially be attributed to RNA polymerase). Any choice for this parameter must necessarily reflect a balance of risks, where lower values (close to 50%, which would roughly match a direct linear fit to the IPOD versus ChIP data points) increase the odds of identifying spurious protein occupancy peaks actually attributable to RNA polymerase, and higher values (up to

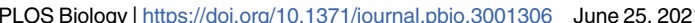

100% would be conceivably justifiable) risk erasing true regulatory protein occupancy due to overzealous RNA polymerase occupancy removal.

We refer to the resulting ChIP-subtracted IPOD signal as the IPOD-HR signal; for analysis and display, we further standardize the signal by calculating robust z scores, where the robust z-score $z_i$ at position i is defined as

$$z_i = (x_i - \text{median}(\mathbf{X}))/\text{mad}(\mathbf{X})$$

for a IPOD-HR data vector $\mathbf{X}$, and mad() indicates the median absolute deviation. In many cases a more useful signal for visualization is a *p*-value for enrichment at each site; $\log_{10}$ *p*-values are calculated under the null hypothesis that the distribution of the robust z-scores is standard normal. To provide uncertainty estimates grounded in observed levels of biological variability across replicates, for each data point, we also constructed an interval between the lowest and highest values that could have been obtained for our occupancy statistics using any combination of biological replicates (potentially different replicates for the IPOD, ChIP, and input samples to construct the largest possible range). To calculate the error bars shown in **Fig 3A**, we then used parametric bootstrapping to generate confidence intervals for the parameters of interest, assuming that the $\log_{10}p$ scaled occupancy of each TFBS followed a log-normal distribution with a mean of the observed mean and standard deviation of one quarter the range between the highest and lowest replicate-wise values (thus treating the range of the pessimistic replicate-wise possible values as an interval expected to contain approximately 95% of observed points); 95% confidence intervals for the average site-wise occupancies were then calculated from 1,000 bootstrap replicates.

**Feature calling.** To identify peaks in the robust z-scaled IPOD-HR data, we applied continuous wavelet transform (CWT) peak calling [70] (as implemented in the scipy.signal package), with a range of widths from 25 bp to 125 bp (at 5-bp increments, reflecting the expected range of peak sizes given the fragmentation of our input DNA) used to generate the CWT matrix, and refer to peaks based on the minimum signal-to-noise ratio threshold at which they appear as peak calls (note that in the CWT method, a range of widths must be specified as part of the algorithm and gives rise only to a single peak set). To provide quantitative measures of the evidence for different peak calls, separate peak sets were generated at several signal to noise thresholds (a full listing can be observed in the set of thresholds shown in **S2 Data**). Each peak call was padded by 30 bp on each side to define the peak region used in subsequent analysis. We show the performance of a variety of calling thresholds in identifying known TFBSs in **S1 Fig**, and based on those findings, use a threshold of 4 for all other quantitative analysis presented here. However, peak sets at all thresholds are supplied in **S2 Data**. Comparisons were made to data from RegulonDB release 9.4 unless otherwise noted.

EPODs were called using an approach similar to that in [10]: We identified EPOD seed regions as any region at least 1,024 bp in length, over which a 512-bp rolling mean exceeded the overall *k*th percentile of a 256-bp rolling mean across the entire chromosome (in all cases acting on the robust z-scored IPODHR data); we used $k = 90$ for the main EPOD calls made in the text and $k = 75$ for the relaxed version used in threshold analysis. In the case of overlapping seeds, only the one with the highest mean was retained. Seed regions were then expanded in both directions as far as possible while maintaining the mean over the entire EPOD call above the threshold noted above and without crossing any location with a robust z-score $\leq 0$. We note that the lower threshold ($k = 75$) matches the threshold used in the original EPOD definitions from [10] (albeit using a rolling median instead of mean), whereas the more stringent threshold is intended to highlight the most robust regions of large-scale protein occupancy; both thresholds show similar properties in terms of overlaps with gene sets and genomic features.

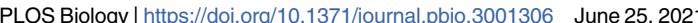

**Transcription factor co-clustering analysis.** For the TF occupancy and co-clustering data, we performed consensus clustering (inspired by [71]). For each biological condition, we assigned each TF a score given by the geometric mean of the site-level occupancies (IPOD-HR $-\log_{10}$ $p$-values) for annotated binding sites of that TF in that condition (using a minimum value of 0.01 for each site-level value); the condition-wise average occupancies for each TF were then divided by the highest average occupancy for that TF across all conditions, yielding an occupancy score on the interval (0,1] for each TF condition combination. The occupancy profiles of TFs across conditions were clustered 100 times using K-means clustering at each number of clusters between 8 and 12 (inclusive); the "co-clustering frequency" κ is defined as the fraction of those 100 trials in which a given pair of TFs were assigned to the same cluster. We then used the quantity (1-κ) as a distance measure in a final hierarchical clustering, assigning the TFs to 10 clusters, to provide the cluster identities shown in **Fig 3C**.

**RNA sequencing and differential expression calling.** RNA-seq data sets were subjected to the same initial preprocessing and quality control steps as outlined above for the DNA samples, and then gene-level expression was quantified using kallisto v0.43 [72] on a version of the MG1655 (GenBank NC_000913) genome with all ribosomal RNAs removed. Gene-level TPM values from kallisto were used for all downstream analysis unless otherwise noted. To generate high-resolution occupancy plots, reads were instead aligned with bowtie2 as described above for DNA reads, and read occupancies quantified using the genomecov command of bedtools2 [73].

## TFBS comparison

Binding sites identified from IPOD-HR peaks (as described above in the "Feature calling" paragraph) were cross-referenced with known and predicted TFBSs using bedtools2 [73]. "Known" sites comprise all binding sites contained in the RegulonDB release 9.4 BindingSite-Set.txt file [7]; "predicted" sites are identified by scanning the MG1655 (GenBank NC_000913) genome with FIMO [74] using all *E. coli* position weight matrices from SwissRegulon [75] (as distributed by the MEME project); each PWM was applied separately, and all sites with a q-value less than 0.2 were retained. Default settings were used for FIMO, except that the background was a second-order Markov model based on the NC_000913 genome, and the number of maximum stored scores was set to 10,000,000.

## Motif identification

Novel sequence motifs implied by IPOD-HR data were identified using an inference pipeline built off of FIRE [35]. Occupancy peaks and associated discrete threshold scores in the IPOD-HR traces were called using the CWT-based approach described above, using a score threshold of 4; each peak was assigned a discrete score corresponding to the average IPOD-HR occupancy score within that peak, rounding down. We then generated a background distribution of unbound sequences drawn from the portion of the genome not included in peaks, matching the length distribution of the peaks but with 3 times as many locations; all such background regions were assigned a score of 0 to distinguish them from the various thresholded peak regions.

Motifs were called using FIRE with 2 separate variations: FIRE_gapped (with parameters—kungapped = 6—gap = 0–10—jn_t_gapped = 4—minr = 0.5), which searches for gapped motifs typical of prokaryotic TFBSs; and FIRE_maxdeg (with parameters—jn_t = 8—minr = 1.5—maxdeg = 1.8), which searches for motifs while preserving information content above a specified threshold. We applied additional empirical filters to specifically enrich for peaks corresponding to binding sites: All peaks identified via FIRE were required to have the motif

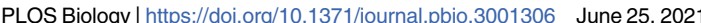

significantly depleted from the background population ($p < 0.01$). To assess the FDR of our methods, we also generated 20 decoy peak sets by shuffling the locations of the real peaks observed in each condition, along with corresponding randomized unbound sets for each, and then applied identical peak calling procedures to each decoy set.

To avoid repeated reporting of very similar motifs which might be identified by our pipelines, we applied the matrix-clustering module of RSAT [76] (using recommended thresholds -lth cor 0.7 -lth w 5 -lth Ncor 0.4) to obtain nonredundant motif sets for downstream analysis. We compared all called motifs with previously known motifs from the SwissRegulon database using TOMTOM [36] with default parameters, requiring an E-value of 0.5 or lower for "Identified" hits. For the identification of predicted regulons associated with each motif, we applied the FIMO program [74] to identify potential binding sites on the *E. coli* K12 genome, with a q-value threshold of 0.2; these were referred to as "strict" motif hits. For the subset of predicted motifs that yielded no potential binding sites at this threshold, we instead report all locations that correspond to the FIMO score of the best single location found in the genome, referring to these as "loose" motif hits. For the purposes of our analysis of the potential regulatory networks of novel motifs), we marked each transcriptional unit in *E. coli* as being regulated by a particular motif if and only if a predicted binding site for that motif was within 100 bp of the annotated transcription start site (in RegulonDB) for that transcriptional unit.

## In vitro pulldown of unidentified transcription factors

In order to identify the protein(s) binding to the *sdaC* promoter (as in **Fig 4**), we first prepared biotinylated bait DNA by cloning a fragment of the *sdaC* promoter (running from positions 2927790 to 2927975 in the U00096.3 genome) into a pAZ3-based cloning vector [77] and then amplifying that region of the plasmid using a primer pair where one primer contained a 5′ biotinylation. The resulting 486-bp fragment was treated with Exonuclease I (Affymetrix) according to the manufacturer's instructions to remove unreacted primer and then purified using a Zymo Clean & Concentrate 25 kit.

The biotinylated bait DNA was then bound to equilibrated Dynabeads MyOne Streptavidin C1 beads (Invitrogen). Beads were equilibrated by washing 3 times with 1x B&W buffer (5 mM Tris Cl, pH 7.5; 0.5 mM EDTA; 1 M NaCl) and then resuspended in 5 volumes of 2x B&W buffer, using 42 μL of the original resuspended bead solution per reaction. The equilibrated beads were combined with 8 μg of biotinylated bait DNA plus an appropriate volume of water to yield a final 1x B&W solution and incubated 15 minutes at room temperature with gentle rocking to allow for bait binding. The beads were then washed 3 times with 500 μL of 1x B&W buffer, twice in 500 μL of 1x BMg/THS buffer (5 mM HEPES, pH 7.5; 5 mM MgCl2; 50 mM KCl; 31 mM NaCl; 1x cOmplete EDTA-free protease inhibitors (Roche)), and once with 500 μL of 1x BMg/THS/EP buffer (1x BMg/THS buffer supplemented with 20 mM EGTA (pH 8.0) and 10 μg/mL poly d(IC) (Sigma)). The beads were then resuspended in 200 μL of BMg/THS/EP buffer and gently mixed by hand for 1 minute to complete equilibration.

Cell extracts were prepared by growing to an OD600 of 0.2 in M9/RDM/glucose media (following the same procedures as those given for IPOD-HR experiments). Once reaching the target OD, the cells were chilled 10 minutes on ice and then pelleted by spinning for 10 minutes at 5,500 x g while at 4˚C. Supernatant was removed, and the cells were flash-frozen in a dry ice/ethanol bath. Cells were then lysed by resuspending the frozen pellet resulting from 82 mL of culture in 160 μL of B-PER II bacterial protein extraction reagent (Thermo Fisher Scientific). We then added 3.8 mL of 1x BMg/THS buffer and 0.8 μL of ReadyLyse lysozyme solution (Lucigen), 40 μL of 10 mg/mL RNase A, 20 μL of CaCl$_2$, and 200 μL of micrococcal nuclease (NEB; 2,000,000 gel units/mL). The lysis/digestion reaction was allowed to proceed

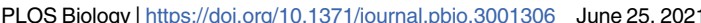

for 30 minutes at room temperature and then clarified by centrifugation at 30 minutes at 16,100x g held at 4˚C. The reaction was halted by the addition of 444 μL of 5 M NaCl, and then the entire volume applied to a 3 kDa MWCO spin filter (Amicon Ultra; Millipore) and centrifuged at (3,200 x g held at 4˚C) until approximately 400 μL of retentate remained. We then added 3.6 mL of BMg/THS (lacking NaCl and KCl, but containing 5 mM CaCl2) and filtered to 400 μL of retentate. Retained liquid was then recovered and diluted to a final volume of 4.0 mL with addition of salt-free BMg/THS + 5 mM $CaCl_2$. The retained lysate was then incubated 30 minutes at room temperature to permit further activity of micrococcal nuclease on remaining DNA in the sample and then quenched with 168 μL of 500 mM EGTA. The volume of the sample was reduced to approximately 1.6 mL by ultrafiltration as above and further supplemented with 10 μg/mL of poly d(IC) and 1 mM dithiothreitol.

Probing of the lysates was then accomplished by combining the equilibrated bait-bead complexes (described above) with the lysates and incubating 30 minutes with rocking at room temperature. The supernatant was then removed, and the beads washed twice with 200 μL of BMg/THS/20 mM EGTA/10 μg/mL poly d(IC) and once with 200 μL of BMg/THS/20 mM EGTA. Proteins were then eluted from the beads through progressive washes of elution buffer (25 mM Tris HCl, pH 7.5) with 100 mM NaCl, 200 mM NaCl, 400 mM NaCl, and 1 M NaCl, with 50 μL used for each elution.

We successively probed the lysates described here with probes containing promoter sequences from *lexA*, *purR*, and finally, *sdaC* (each containing identical plasmid-derived flanking sequences). An approximately 25-kDa band of interest appeared in the 400 mM and 1 M NaCl *sdaC* eluates but not eluates from a parallel experiment performed under identical conditions with a segment of the *thiC* promoter; these bands were excised from a silver-stained gel. The 400 mM gel slice was then subjected to proteomic analysis at the University of Michigan Proteomics & Peptide Synthesis core facility. The gel slice was processed using a ProGest robot (DigiLab) to wash with 25 mM ammonium bicarbonate followed by acetonitrile, reduce with 10 mM dithiothreitol at 60˚C followed by alkylation with 50 mM iodoacetamide at room temperature, digested with trypsin (Promega) at 37˚C for 4 hours, and then quenched with formic acid. The digest was then analyzed by nano liquid chromatography with tandem mass spectrometry (LC-MS/MS) with a Waters NanoAcquity HPLC system interfaced to a Thermo Fisher Q Exactive. Peptides were loaded on a trapping column and eluted over a 75-μm analytical column at 350 nL/min; both columns were packed with Jupiter Proteo resin (Phenomenex). The injection volume was 30 μL. The mass spectrometer was operated in data-dependent mode, with the Orbitrap operating at 60,000 full width at half maximum (FWHM) and 17,500 FWHM for mass spectrometry (MS) and tandem mass spectrometry (MS/MS), respectively. The 15 most abundant ions were selected for MS/MS. Data were searched using a local copy of Mascot, and Mascot DAT files were parsed into the Scaffold software for validation, filtering, and to create a nonredundant list per sample. Data were filtered using 90% protein and 95% peptide probability thresholds (Prophet scores) and requiring at least 2 unique peptides per protein. The resulting mass spectrometry analysis is given in **S3 Table** after manual pruning by core staff of common contaminants (e.g., human keratin).

## Miller assay

Cells containing the *lacZ* reporter controlled by *sdaC* promoter variants (see strain construction notes above) were grown overnight in M9 RDM media with 0.4% glucose. In the morning, cells were diluted 1:300 in fresh, prewarmed M9 RDM media with 0.4% glucose in 96-well plate. The plate was placed in a BioTek Synergy H1 plate reader measuring OD600 and grown until mid-exponential phase, around 6 doublings. Once the cells reached the target OD of

around 0.2 (1-cm path length equivalent), 80 uL of cells were placed in a new 96-well plate and mixed with 120 uL of β-galactosidase assay mix (see below), avoiding the production of bubbles. The plate was loaded into the same Synergy H1 plate reader and incubated at 37˚C, taking measurements every 2 minutes for 1 hour. Our procedure, assay, and analysis for the Miller assay are adapted from [78].

Reagents:

- Z-buffer: 60 mM $Na_2HPO_4$, 40 mM $NaH_2PO_4$, 10 mM KCl, 1 mM $MgSO_4$

- β-mercaptoethanol solution: 2.7 μl/mL β-mercaptoethanol in Z-Buffer

- ONPG solution: 4 mg/mL ONPG in Z-Buffer

- Lysozyme solution: 10 mg/mL lysozyme in 10 mM Tris-HCl, pH = 8.0

    β-galactosidase assay mix (100 reactions):

- 8 mL β-me solution

- 3 mL ONPG solution

- 800 μl PopCulture Reagent (Sigma-Aldrich)

- 200 μl Lysozyme solution.

We then calculated the enzymatic activity in each well first by subtracting 1.7 times the OD550 value at each time point from the OD420 value (to yield the amount of enzymatic product after correcting for cell debris) and then fitting a LOESS smoothing curve to the resulting values and identifying the point of maximal slope; that slope (which is a rate of increase in product versus time) is taken as the enzymatic activity. The background activity of MG1655 *lacZ*::*cml* cells measured at the same time was then subtracted from each data point. The activities were then divided by the OD600 of the corresponding cells used in each assay to account for the differing amounts of biological starting material to yield final OD-normalized Miller units. The experiments were performed in technical duplicate on each of 4 different days.

The resulting normalized Miller values were then analyzed using a Bayesian mixed-effects model to identify the relative promoter activities in our 8 strains of interest. We assumed that the normalized Miller values followed the form

$$M_{cent} \sim t(\alpha_{date} + \beta_{strain} + \gamma_{date:strain}, \ \sigma, \ \nu),$$

where α and β are fixed effects representing the effects of experimental date and of the *yieP*/ promoter genotypes, respectively; the γ is a random effects term with levels for each date/strain combination, and σ and ν are the standard deviation and degrees-of-freedom parameters of the t distributed residual noise. $M_{cent}$ indicates centered Miller units where the global mean of all measurements was subtracted from each value (the fitted coefficients were then decentered prior to plotting). The model was fitted using the brms [79] module of R; using a normal (0,1,000) prior on each of the fixed effects, a t (0, 500, 10) prior on the standard deviation parameter and defaults for all other parameters (the mildly informative priors used here were based on the behavior of WT MG1655 cells induced by IPTG in our assay). Model convergence was assessed using the Rhat metric and manual inspection of the posterior predictive distribution.

## Supporting information

**S1 Text. Additional information on the effects of RNA polymerase on large-scale protein occupancy.** See also **S5**–**S8** Figs.
(PDF)

**S2 Text. Additional information on the recovery of known TFBS profiles from IPOD-HR data and inferred motifs.** IPOD-HR, in vivo protein occupancy display—high resolution; TFBS, transcription factor binding site.
(PDF)

**S1 Fig. Effect of peak calling threshold on coverage and enrichment of known TFBSs.** Data are shown for the WT cells in the RDM condition. Shown is the fraction of the entire genome contained in peak calls (left vertical axis, blue line) or the enrichment of TFBSs overlapping those peak calls relative to that expected by chance (right vertical axis, red line). Overlaps at all shown thresholds were statistically significant ($p < 0.01$, permutation test in each case). RDM, rich defined medium; TFBS, transcription factor binding site; WT, wild-type.
(PNG)

**S2 Fig. Average occupancies for TFs with known/characterized binding sites. (A)** Shown for each TF (row) is the geometric mean of site-level occupancies for all detectable sites for that TF under that condition. "Detectable sites" refer to RegulonDB-annotated sites which had a robust z-score of at least 3 under at least 1 condition; factors with fewer than 3 detectable sites were excluded. The values within a single site, for a single condition, are summarized by the maximum occupancy within that site, reflecting the peak of the observed binding signal. The TFs are ordered based on a consensus clustering approach as applied for **Fig 3** of the main text. Raw data on the underlying site-level occupancies are given in **S8 Data**. **(B)** As in **A**, subsequently scaling each row by its maximum value so the highest occupancy condition for each TF receives a score of 1.0. TF, transcription factor.
(PNG)

**S3 Fig. Interplay of H-NS occupancy, EPOD locations, and transcription. (A)** Mean levels of H-NS binding (data from [44]) for all EPODs called in the WT rich media condition; each point shows either an EPOD or a single contiguous non-EPOD region. Each point is colored by its classification into high, medium, or low H-NS binding using a Gaussian mixture model with 3 groups, after removal of outliers using the local outlier factor [80] as implemented in the python scikit-learn module [81], using 25 neighbors and default settings for other parameters. **(B)** Distributions of mean RNA read density stratified by the H-NS binding categories shown in panel **A**, with each case divided by EPOD status. The median of each group is shown by a dashed line and the 25th and 75th quartiles by dotted lines. "*" indicates a significant difference between the EPOD vs. background groups ($p < 0.05$, Mann–Whitney U test). EPOD, extended protein occupancy domain; WT, wild-type.
(PNG)

**S4 Fig. Identification of RNA polymerase vs. non-RNA polymerase protein occupancy.** Shown is a density plot of the $\log_2$(IPOD/Input) signal vs. $\log_2$(RNA polymerase ChIP/Input) signal, demonstrating the presence of 3 subpopulations of genomic positions: unbound positions (without enrichment using either protein occupancy profiling method), RNA polymerase occupancy (part of a highly correlated region of high IPOD occupancy and high RNA polymerase occupancy), and occupancy with other proteins (which shows high IPOD occupancy but low RNA polymerase occupancy). Note that there is no corresponding population of high RNA polymerase occupancy but low IPOD occupancy, rather, the RNA polymerase-

bound regions are a subset of the regions detected by IPOD. Color intensity scales logarithmically with bin occupancy. ChIP, chromatin immunoprecipitation.
(PNG)

**S5 Fig. Overlaps of EPOD sets resulting from different calling methods.** Shown in the heat map are the fraction of EPODs from the EPOD set defined by the row label that overlap the EPOD set defined by the column label. Asterisks reflect *p*-values arising from a Monte Carlo permutation test (1,000 random circular permutations of the EPOD locations; * $p < 0.05$, ** $p < 0.01$, *** $p < 0.001$). *p*-Values for the overlaps between the +RIF IPOD-HR EPOD set and the Vora heEPODs were >0.8 for both directions of comparisons; a full list of values is given in S2 Table. EPOD, extended protein occupancy domain; heEPOD, highly expressed extended protein occupancy domain; IPOD-HR, in vivo protein occupancy display—high resolution.
(PNG)

**S6 Fig. Effects of rifampin treatment on protein occupancy of a highly transcribed region.** Shown are occupancy signals for interphase-extracted, RNA polymerase ChIP, and ChIP-subtracted IPOD occupancy (IPOD-HR) samples in the vicinity of a large cluster of ribosomal protein genes (running from *rplQ* to *rpsJ*). Signals are log2 extracted:input ratios (for IPOD and ChIP samples) or ChIP-subtracted robust z scores (IPOD-HR). ChIP, chromatin immunoprecipitation; IPOD-HR, in vivo protein occupancy display—high resolution.
(PNG)

**S7 Fig. Effects of rifampin treatment on protein occupancy of a transcriptionally silent region.** Shown are occupancy signals for interphase-extracted, RNA polymerase ChIP, and ChIP-subtracted IPOD occupancy (IPOD-HR) samples in the vicinity of the *waaQGPSBO-JYZU* operon, which was identified as a strong tsEPOD in [10]. Signals are log2 extracted:input ratios (for IPOD and ChIP samples) or ChIP-subtracted robust z scores (IPOD-HR). ChIP, chromatin immunoprecipitation; IPOD-HR, in vivo protein occupancy display—high resolution; tsEPOD, transcriptionally silent extended protein occupancy domain.
(PNG)

**S8 Fig. Overlaps of inferred motifs with binding sites for TFs with similar actual motifs.** **(A)** Violin plots showing the $\log_{10}$-fold enrichment (or depletion) of overlap between the indicated motif-based binding site calls (using IPOD-HR inferred motifs or motifs from SwissRegulon) with annotated binding sites from RegulonDB; matches of inferred motifs with TFs arise from TOMTOM calls (see text for details). A pseudocount of 0.0001 is added to each overlap. For motif hits in the present figure, "loose" motif hits were used if strict hits were not available (see Methods for details). **(B)** $\log_{10}$ recall for identification of annotated (from RegulonDB) binding sites for each set of motif-based calls indicated in panel **A**; a pseudocount of 0.0001 is added to each value to avoid singularities. **(C)** Recall of annotated sites for each indicated TF (matching those shown in panels **A** and **B**) using either the union of all peaks called from our IPOD-HR data set at a peak calling threshold of 4 ("combined peaks") or the union of all binding sites for our nonredundant motif set ("combined motifs"). Red dashed lines show the fraction of the genome covered by the peaks and motifs (depending on the axis), and thus represent the recalls that would be expected solely by chance. IPOD-HR, in vivo protein occupancy display—high resolution; TF, transcription factor.
(PNG)

**S1 Table. Comparisons of the distributions of robust z-scores (without log scaling) observed on different genomic regions in the "WT,rich" condition; see also cumulative distributions of the $\log_{10}p$ statistic in Fig 2E.** The higher means and rightward skews of all other

data sets relative to the "Coding, No TF" portion indicates that the portions of the genome with higher IPOD-HR robust z-scores are associated with noncoding regions and annotated TFBSs. The "vs. Coding No TF" column gives the *p*-value for a permutation test comparing the mean z-scores in the indicated genomic region with those in the "Coding, No TF" region; the permutation test was conducted using 200 random rotations of the data values relative to the feature coordinates, ensuring that the correlation structures of both data and features were conserved. IPOD-HR, in vivo protein occupancy display—high resolution; TF, transcription factor; TFBS, transcription factor binding site; WT, wild-type.
(PDF)

**S2 Table. *p*-Values arising from the statistical tests shown in Fig 3C (calculated using iPAGE software) and in S4 Fig (arising from permutation tests in which the genomic coordinates of 1 EPOD set were rotated in unison relative to the other, to preserve the internal correlation structures of each group).** EPOD, extended protein occupancy domain.
(XLSX)

**S3 Table. Mass spectrometry identified peptide counts showing abundances of proteins pulled down by biotinylated bait DNA from the *sdaC* promoter region, after pruning of likely contaminants (see Methods for details).**
(PDF)

**S4 Table. Summary of EPOD characteristics across experimental conditions.** The "Median difference" column refers to the difference in median robust Z-scores between EPODs and all other sites in the genome, with positive values indicating higher levels within EPODs. *p*-Values for a significant difference are obtained using a resampling test, with 1,000 random circular permutations of the EPOD locations on the genome (thus preserving the correlation structure of genomic features); q-values are obtained by correction of the *p*-values using the Benjamini–Hochberg method [82]. All sequence features were subjected to a 500-bp rolling mean prior to overlap calculation. EPOD, extended protein occupancy domain.
(PDF)

**S5 Table. List of IPOD-HR experiments performed over the course of the study, including concise names used to refer to each experiment in the remainder of the text (included as a separate file).** Note that "Aligned reads" refers specifically to the number of concordant, uniquely aligned read pairs arising from a given sample, or the number of pseudoaligned reads in the case of an RNA sample. IPOD-HR, in vivo protein occupancy display—high resolution.
(XLSX)

**S1 Data. Bedgraph files showing the processed occupancy traces for the biological conditions considered in this study, corresponding to the plotted data.**
(ZIP)

**S2 Data. GFF files containing the locations of all peak calls (with cutoffs indicated in the "score" field; see Methods for details) obtained from our IPOD-HR occupancy profiles through all conditions in the present study.** The value in the "score" column corresponds to the peak calling threshold in use. Note that the "strand" and "frame" fields convey no useful information. IPOD-HR, in vivo protein occupancy display—high resolution.
(ZIP)

**S3 Data MEME-formatted file containing the complete set of redundancy-pruned motifs discovered across all conditions in the present study.**
(ZIP)

**S4 Data. GFF file giving the FIMO hits on the *E. coli* MG1655 genome for all newly inferred motifs described here.** For factors that gave hits using our standard criterion (FIMO q-value <0.2), we flag hits as "Nucleotide motif (strict)"; for motifs that gave no hits according to this criterion, we instead report all locations that correspond to the FIMO score of the best single location found in the genome, flagged as "Nucleotide motif (loose)".
(ZIP)

**S5 Data. Table showing the Jaccard index between predicted regulons of each of the newly inferred motifs, with all TFs that have characterized regulons in RegulonDB.** Jaccard indices were calculated at the gene level (using gene names as individual items), using the definitions of regulated genes used elsewhere in the present paper for the newly inferred motifs and the network_tf_gene.txt entries from RegulonDB (release 10.8). Motif indices match those in S3 Data for members of the redundancy-pruned set. TF, transcription factor.
(ZIP)

**S6 Data. Table showing the GO terms that are enriched in the potential regulon of each newly inferred regulatory motif.** The tab-separated table gives the motif name, GO term, and *p*-value (obtained from iPAGE) for each inferred regulatory connection. GO, gene ontology.
(ZIP)

**S7 Data. GFF file containing the locations of all EPODs called on our IPOD-HR occupancy profiles (see Methods for details).** Note that the "score," "strand," and "frame" fields convey no useful information. EPOD, extended protein occupancy domain; IPOD-HR, in vivo protein occupancy display—high resolution.
(ZIP)

**S8 Data. Table showing the occupancy at each detectable TFBS used in the analysis of Fig 3 and associated Supporting information figures.** TFBS, transcription factor binding site.
(ZIP)

## Acknowledgments

We are grateful to Dr. Alison Hottes for technical assistance and suggestions on the original IPOD method and to Ms. Christine Ziegler for optimization of the Miller assay as used in Fig 4.

## Author Contributions

**Conceptualization:** Saeed Tavazoie.

**Data curation:** Lydia Freddolino, Saeed Tavazoie.

**Formal analysis:** Lydia Freddolino, Saeed Tavazoie.

**Funding acquisition:** Lydia Freddolino, Saeed Tavazoie.

**Investigation:** Lydia Freddolino, Haley M. Amemiya, Thomas J. Goss.

**Methodology:** Lydia Freddolino, Saeed Tavazoie.

**Software:** Lydia Freddolino.

**Writing – original draft:** Lydia Freddolino, Saeed Tavazoie.

**Writing – review & editing:** Lydia Freddolino, Haley M. Amemiya, Saeed Tavazoie.

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

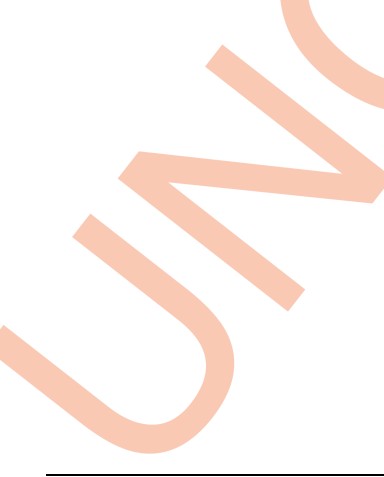

*tsx*-p2 promoter of *Escherichia coli* K-12. J Bacteriol. 1991; 173:5419–30. https://doi.org/10.1128/jb. 173.17.5419-5430.1991 PMID: 1715855

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

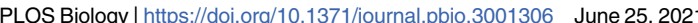