## [Editor Report · Decision Letter 0]

9 Nov 2020

Dear Dr. Tavazoie, 

Thank you for submitting your manuscript entitled "Dynamic landscape of protein occupancy across the Escherichia coli chromosome" for consideration as a Research Article by PLOS Biology.

Your manuscript, the reviews and your revision plan have now been evaluated by the PLOS Biology editorial staff, as well as by an academic editor with relevant expertise, and I'm writing to let you know that we would like to consider your manuscript further.

However, before we can send a "Major Revision" decision, we need you to complete your submission by providing the metadata that is required for full assessment. To this end, please login to Editorial Manager where you will find the paper in the 'Submissions Needing Revisions' folder on your homepage. Please click 'Revise Submission' from the Action Links and complete all additional questions in the submission questionnaire.

Please re-submit your manuscript within two working days, i.e. by Nov 11 2020 11:59PM.

Once your full submission is complete, your paper will undergo a series of routine checks. Once your manuscript has passed all checks I will be able to send you your formal decision. 

Kind regards,

Paula

---

Associate Editor

PLOS Biology

---

## [Editor Report · Decision Letter 1]

11 Nov 2020

Dear Dr. Tavazoie,

Thank you very much for submitting your manuscript "Dynamic landscape of protein occupancy across the Escherichia coli chromosome" for consideration as a Methods and Resources at PLOS Biology. Your manuscript and the three Review Commons reviews has been evaluated by the PLOS Biology editors, and by an Academic Editor with relevant expertise.

In light of the reviews (below), we will not be able to accept the current version of the manuscript, but we would welcome re-submission of a much-revised version that takes into account the reviewers' comments. We cannot make any decision about publication until we have seen the revised manuscript and your response to the reviewers' comments. Your revised manuscript is also likely to be sent for further evaluation by the reviewers.

We expect to receive your revised manuscript within 3 months. 

In particular, referee #1 thinks that a user-friendly accessible browser format is really needed, asks about promoters with more than one protein bound, asks whether it is possible to substract H-NS ChIP signals, and says that you should make a clear statement about what is missing. Referee #2's main concern is the source code is missing. Referee #2 also thinks that you should re-write the description of protein occupancy and peak calling in the method section for clarity and asks specific questions about it, says that the functional validation of YieP binding site is not convincing enough. Referee #3 has several suggestions regarding the integration and cross referencing of your findings with existing data, and has questions about TF and co-regulation. Taking these comments into account, we think that your plan for revision is reasonable.

**IMPORTANT - SUBMITTING YOUR REVISION**

*Re-submission Checklist*

*Published Peer Review*

*PLOS Data Policy*

*Blot and Gel Data Policy*

Sincerely,

Paula

---

Associate Editor,

pjaureguionieva@plos.org,

PLOS Biology

REVIEWS (From Review Commons):

Reviewer #1:

The paper describes a technique that produces an IPOD (in vivo protein occupancy display)

across the entire E coli chromosome. By subtracting out signals due to RNA polymerase,

this display is HR (high resolution) in the sense that the authors can see signals due to

individually bound proteins. The drawback of the method is that the signals seen in the

IPOD are anonymous, but the authors describe in some detail how the display can be

exploited. The paper is clearly written and presented, and will be accessible to students,

with some effort, and easy to grasp for those who have followed the topic, especially those

familiar with the previous 2009 Mol Cell Vora et al paper.

The key conclusions are convincing. Of course the methodology has inbuilt issues, like all

top-down methods that rely on a particular probe, and these will bias the results, but the

authors assess the pros- and cons realistically. Concerning the data, I would have liked to

see more of the IPOD in the format in Figure 2. I guess this is somewhere in the

Supplementary data, but I was unable easily to access it. What is really needed is a userfriendly accessible browser format.

The authors make a convincing case that certain 'spikes' in the IPOD correspond to

individual proteins bound. So what about promoter regions where several proteins are

bound e.g. the lac operon regulatory region where one might expect to see Lac Repressor

and CAP? Or the galETK promoter where there could be Gal Repressor, HU and CAP?

Readers might like to check their favorite promoters for themselves. At the very least, the

authors need to make a clear statement about what is missing.

For me the highlight was the observation that EPODs (extended protein occupancy

domains) don't change much according to growth conditions. This leads to a very nice view

of proteins on the E coli 'landscape' where regulatory features vary but EPODs don't (well,

not much, anyway). The non-HNS EPODS are intriguing. Is it possible to do a subtraction

of H-NS ChIP signals as was done for RNAP ChIP signals?

Significance

The opening paragraph of the discussion explains the relative merits of bottom-up versus

top-down approaches to understanding bacterial regulatory networks, and carefully

explains that IPOD-HR is an addition to the top-down told. This work is of interest to those

concerned with bacterial chromosome organisation and the regulation of bacterial gene

expression.

Reviewer #2:

The authors describe a novel approach that allows characterization of the protein-DNA

interaction dynamics at the global scale. The method is a further advance on protein

occupancy display (IPOD) developed earlier by the same authors. The crucial modification

introduced in the present work is the use of rifampicin treatment to distinguish the

contribution from sequence-specific transcription factors and RNA polymerase. The new

method is referred to as IPOD-HR (for high resolution).

They apply this method across different physiological conditions (exponential and

stationary phase, rich or minimal medium) and validate the results using a set of well

characterized transcriptional regulators (ArgR, LexA, PurR) and corresponding mutants.

They then utilize an unsupervised clustering algorithm on the protein occupancy data for

annotated transcription factors and show that transcription regulators belonging to the same

GO term tend to cluster together across the conditions.

As a large number of protein occupancy peaks obtained in this study does not correspond to

the annotated transcription factor binding sites in promoter, the authors select one such site

and use it to fish out and identify the interacting protein (YieP). They validate the

functional importance of this interaction by in vitro DNA-binding assays and RNA-Seq

data for the corresponding mutant which show a significant reduction of the regulated

transcript (sdaC) level.

Building on this result the authors examine the sequences corresponding to IPOD-HR

peaks and find that about a half of them do not match any known (or predicted)

transcription factor binding sites. They use computational motif discovery algorithm FIRE

to IPOD-HR data to find about 200 novel motifs. These account for almost a third of the

observed protein occupancy peaks. Importantly, the inferred motifs show significant

enrichment in promoter but not ORF regions.

The last part of this work explores so-called extended protein occupancy domains (EPODs)

which were first reported by the authors in the previous work and apparently correspond to

kilobase-scale regions of high level of protein occupancy. The correction for RNA

polymerase occupancy levels introduced in the present study allowed the authors to claim

that most EPODs are transcriptionally silent, akin to eukaryotic heterochromatin regions.

One striking result is that EPODs appear to be almost invariant across the range of

physiological conditions used in this study. The pathways enriched in EPODs regions

(prophage genes and mobile elements) as well as the strong overlap with H-NS ChIP-Seq

data prompt the authors to suggest that the main role of EPODs is the transcriptional

silencing of xenogenetic material. Nonetheless, a significant fraction of EPODs does not

overlap with H-NS binding and provide an intriguing opportunity for further study.

The strength of the method is that no prior knowledge of the DNA interacting protein is

required and it can be used to survey the protein occupancy of both transcription regulatory

sites and large-scale protein-bound domains of bacterial chromosome. This will be a

valuable addition to the existing toolkit of high-throughput methods and has a great

potential when applied to other organisms and physiological conditions.

However, several points need to be addressed before the work can be published.

1.My main concern is that being heavily dependent on computational analysis this study 

does not provide the source code for the analysis performed, specifically protein occupancy

and feature (peak) calling and transcription factor co-clustering. While I agree that the

inclusion of code for routine steps in NGS data processing such as QC and alignment adds

little value to the reader, these are computational steps on which the entire method is built

and they should be included in the manuscript as a link to the source code.

2.The description of protein occupancy and peak calling in the method section is difficult to

parse and probably should be re-written for better clarity. Specific questions that arise

while reading this part:

- Why is spline used for smoothing abundances? The choice of smoothing function and its

parameters needs better explanation. The same goes for the choice of LOESS fit for RNA

polymerase normalization.

- Where did 1.1 scaling factor come from?

- What's the effect of data downsampling on the fit robustness?

- Similarly, the selection of parameters used in peak calling in both CWT and EPODs cases

needs to be explained. They seem arbitrary without additional clarification.

3.The functional validation of YieP binding site is not convincing enough. The transcript

level depletion in mutant RNA-Seq data might be an indirect effect. A reporter gene assay

will provide a more direct evidence that the site is indeed a functional transcriptional

regulator.

4.Perhaps outside of the scope of this study but it would be great to provide some kind of

functional validation to the novel transcription factor motifs reported here.

Reviewer #3:

**Summary:**

In this manuscript, Freddolino and collaborators introduced their improved in vivo protein

occupancy display technology (IPOD-HR) for genome-wide profiling of protein

occupancy. The IPOD-HR workflow integrates protein occupancy data and RNA

polymerase ChIP-seq data to identify all regions of a bacterial genome occupied by DNAbinding proteins. Using the IPOD-HR technology, the authors identified ~19,000 potential

TF-DNA interactions in the genome of Escherichia coli across six different

conditions/genotypes. Remarkably, protein occupancy profiles generated using IPOD-HR

recapitulated a high proportion of previously reported TF binding sites in E. coli

chromosome, identified ~90% of the known DNA binding motifs for E. coli TFs, and

discovered thousands of novel TF binding sites that await further characterization. The

authors showed that protein occupancy profiles (generated using IPOD-HR) in combination

with prior knowledge about a transcriptional regulatory network (TRN) of interest may

offer valuable information about the regulatory activity of TFs, uncharacterized gene

promoters, and binding affinity of uncharacterized putative TFs. Therefore, this new

technology may facilitate the reconstruction of condition-specific bacterial TRNs.

**Major comments:**

The authors have done an excellent job showing, with specific examples, potential

applications of their new technology. However, the manuscript would deeply benefit from a

more comprehensive analysis of the functional implications of the TF-DNA binding events 

discovered using IPOD-HR. Authors should use the protein occupancy profiles they have

already generated to reconstruct the binding profiles of individual regulators across

different conditions (at least for global regulators with dozens of targets). Authors may

achieve this goal by genome-wide scanning of the de novo motifs (matched with known TF

binding motifs) in the occupied chromosomal regions or using similar approaches. This

type of analysis is necessary to assess the full potential of their technology and to uncover

the conditional partition of TF regulons, which would offer key insights about the

architecture and dynamic of E. coli TRN. Without this level of resolution, ChIP-seq

experiments would still be required to compare the binding profiles of any TF of interest

across multiple conditions, reducing the value of IPOD-HR. Finally, I encourage the

authors to compare the IPOD-HR reconstructed TF regulons with the information available

in RegulonDB.

**Specific comments**

-L141-148: the authors should clearly explain in 2-3 sentences the main changes in IPODHR with respect to IPOD (RNA polymerase occupancy correction, etc). What are the main

changes in the analytical framework? It is unclear what the authors meant.

-Fig 2E: authors should evaluate the statistical significance of IPOD-HR occupancy.

-Fig 3B: do the TF modules indicate co-regulation? In other words, do TFs that cluster

together share target genes or they only have similar average occupancy profiles?

Moreover, do the authors use any TF regulon size threshold when computing average

occupancy score? If not, how do the authors correct for overlap between regulons (i.e.

between small regulons and large regulons)?

-L331-333 & Supp. Data S1: the authors, if possible, should include information about TFs

potentially binding to the identified peaks. This information will increase the value of the

data generated in this study.

-L360-365: an important piece of information missing in the current version of the

manuscript is the recall rate for known TF regulons (i.e. how many of the known targets of

each TF was identified by IPOD-HR). This is information is important to validate your

results beyond TF motifs. Additionally, did the IPOD-HR data discover novel targets for

previously characterized TFs? If yes, the authors should describe these findings. Finally,

can the authors evaluate if the reconstructed regulons are supported by transcriptional data

(collected in this study) or publicly available data? This would clarify the function of the

putative binding events.

-L397-406: the authors should evaluate the overlap between regulons associated with newly

inferred motifs and previously characterized TF regulons. This comparison may offer

information about the potential role of the uncharacterized TFs and combinatorial

regulation.

-L530-534: It is unclear to me how you would compensate for the lack of prior information

when using the IPOD-HR technology for reconstructing TRNs of uncharacterized species?

**Minor comments:**

-L60-62: and not all binding events are functional. Especially in the case of TF overexpression commonly used in ChIP-seq experiments.

-Fig 3A: authors should indicate how many target genes were used for estimating the

occupancy of each TF. 

-Fig 3C: please add the actual correlation score. Is this anti-correlation a general trend (if

you choose any random genes) or specific for the ArgR and PurR targets?

-L764: please remind the reader why rifampicin is used.

-L828: for how many conditions did the authors perform RNA polymerase ChIP-seq with

samples that were not from the same culture used for IPOD? Can the authors show that

using different samples for IPOD and RNApol ChIP-seq does not change the estimated TF

occupancy profiles?

-L1142: No access to the data (GEO requires a password)

-Table S4: not available.

The reviewer field of expertise is reconstruction of gene regulatory networks.

---

## [Decision Letter · Decision Letter 2]

14 May 2021

Dear Dr. Tavazoie,

Thank you for submitting your revised Methods and Resources entitled "Dynamic landscape of protein occupancy across the Escherichia coli chromosome" for publication in PLOS Biology. I have now obtained advice from two of the original reviewers and have discussed their comments with the Academic Editor. 

Based on the reviews, we will probably accept this manuscript for publication, provided you satisfactorily address the remaining points raised by the reviewers. Please also make sure to address the following data and other policy-related requests.

You will see that reviewer #3 would like you to add a column with the information regarding the closest gene to each motif binding site, and asks why the same genome-wide scan of motif binding sites used to generate Supplementary Data 4 cannot be applied to the non-redundant motifs that matched with known DNA binding motifs for E. coli. Please address the remaining issues from reviewer #3. 

DATA POLICY:

Regardless of the method selected, please ensure that you provide the individual numerical values that underlie the summary data displayed in the following figure panels as they are essential for readers to assess your analysis and to reproduce it: Figures 2B, 2C, 2D, 2E, 2F, 2G, 3A, 3D, 3E, 3F, 3G, 4A, 4E, 5A, 5B, 5C, 5D, 5G, 5H, 6B, 6E, Supplementary figures 3A, and 3B.

We require the original, uncropped and minimally adjusted images supporting all blot and gel results reported in an article's figures or Supporting Information files. We will require these files before a manuscript can be accepted so please prepare and upload them now. Please carefully read our guidelines for how to prepare and upload this data: https://journals.plos.org/plosbiology/s/figures#loc-blot-and-gel-reporting-requirements. We will require this for Figure 4C.

We expect to receive your revised manuscript within two weeks. 

*Published Peer Review History*

*Early Version*

Sincerely,

Paula

---

Associate Editor,

pjaureguionieva@plos.org,

PLOS Biology

Reviewer remarks:

Reviewer #2: Evgeny Nudler. Transcription Elongation and Gene Control.

Reviewer #3: Reconstruction of gene regulatory networks.

Reviewer #2: The authors did a great job addressing every point I made. I support the publication of the manuscript in its present form.

Reviewer #3: The revised manuscript describes the IPOD-HR technology for genome-wide profiling of protein occupancy. The authors comprehensively showed applications of the IPOD-HR method using E. coli as their model system. In this revised version, the authors satisfactorily addressed most of the issues I have previously raised. There are two minor analyses that could benefit the manuscript before acceptance to PLOS Biology:

1. As a method & resource article, would it be possible for authors to add a column with the information regarding the closest gene to each motif binding site listed on Supplementary Data 4? I imagine authors already compiled that information in order to perform the GO enrichment analysis shown in Supplementary Data 5.

2. I understand the authors' response about the current limitations to reconstruct individual TF regulons using a limited set of conditions. However, I still don't understand why the same genome-wide scan of motif binding sites used to generate Supplementary Data 4 cannot be applied to the non-redundant motifs that matched with known DNA binding motifs for E. coli? This could offer potential information about individual (partially characterized) TF regulons. Authors could highlight the caveats they have mentioned so readers are aware of any limitation when interpreting the data.

Minor point:

 3. Fig S2 panel labels are missing. Also, authors referred to TF clusters in their response but those are not shown in the figure S2.

---

## [Editor Report · Decision Letter 3]

2 Jun 2021

Dear Dr. Tavazoie,

On behalf of my colleagues and the Academic Editor, Matthew K. Waldor, I am pleased to say that we can in principle offer to publish your Methods and Resources "Dynamic landscape of protein occupancy across the Escherichia coli chromosome" in PLOS Biology, provided you address any remaining formatting and reporting issues. These will be detailed in an email that will follow this letter and that you will usually receive within 2-3 business days, during which time no action is required from you. Please note that we will not be able to formally accept your manuscript and schedule it for publication until you have made the required changes.

PRESS

Thank you again for supporting Open Access publishing. We look forward to publishing your paper in PLOS Biology. 

Sincerely, 

Paula

---

Paula Jauregui, PhD 

Associate Editor 

PLOS Biology